# *Salmonella* exploits membrane reservoirs for invasion of host cells

Hongxian Zhu[1,2], Andrew M. Sydor ●[1], Kirsten C. Boddy[1,3], Etienne Coyaud ●[4,5], Estelle M. N. Laurent[4,5], Aaron Au ●[6], Joel M. J. Tan ●[1], Bing-Ru Yan ●[1], Jason Moffat ●[2,6,7], Aleixo M. Muise ●[1,8,9,10], Christopher M. Yip ●[6,8], Sergio Grinstein ●[1,3,8], Brian Raught[4,11] & John H. Brumell ●[1,2,3,10] ✉

*Salmonella* utilizes a type 3 secretion system to translocate virulence proteins (effectors) into host cells during infection[1]. The effectors modulate host cell machinery to drive uptake of the bacteria into vacuoles, where they can establish an intracellular replicative niche. A remarkable feature of *Salmonella* invasion is the formation of actin-rich protuberances (ruffles) on the host cell surface that contribute to bacterial uptake. However, the membrane source for ruffle formation and how these bacteria regulate membrane mobilization within host cells remains unclear. Here, we show that *Salmonella* exploits membrane reservoirs for the generation of invasion ruffles. The reservoirs are pre-existing tubular compartments associated with the plasma membrane (PM) and are formed through the activity of RAB10 GTPase. Under normal growth conditions, membrane reservoirs contribute to PM homeostasis and are preloaded with the exocyst subunit EXOC2. During *Salmonella* invasion, the bacterial effectors SipC, SopE2, and SopB recruit exocyst subunits from membrane reservoirs and other cellular compartments, thereby allowing exocyst complex assembly and membrane delivery required for bacterial uptake. Our findings reveal an important role for RAB10 in the establishment of membrane reservoirs and the mechanisms by which *Salmonella* can exploit these compartments during host cell invasion.

*Salmonella enterica* serovar Typhimurium (*S*Tm) is a gram-negative intracellular bacterial pathogen that is a major cause of foodborne gastroenteritis in humans. These bacteria can rapidly invade host cells using a type 3 secretion system (T3SS) encoded by *Salmonella* pathogenicity Island (SPI-1)[1]. SPI-1 T3SS effectors are known to promote the delivery of membrane to the invasion site via the exocyst complex, contributing to the formation of ruffles that envelop

bacteria[2]. However, the membrane source required for exocyst-dependent ruffle formation remains unclear.

Ruffles envelop bacteria, allowing subsequent internalization into *Salmonella*-containing vacuoles (SCVs)[3]. The SPI-1 T3SS effector SopD contributes to host cell invasion by promoting rapid severing (scission) of the plasma membrane (PM) invaginations at invasion sites, thereby generating SCVs[4]. SopD encodes a GTPase activating

[1]Cell Biology Program, Hospital for Sick Children, Toronto, ON, Canada. [2]Department of Molecular Genetics, University of Toronto, Toronto, ON, Canada. [3]Institute of Medical Science, University of Toronto, Toronto, ON, Canada. [4]Princess Margaret Cancer Centre, University Health Network, Toronto, ON, Canada. [5]Protéomique, Réponse Inflammatoire, Spectrométrie de Masse (PRISM)-U1192, Université de Lille, Inserm, CHU Lille, Lille, France. [6]Institute of Biomedical Engineering, University of Toronto, Toronto, ON, Canada. [7]Genetics and Genome Biology Program, Hospital for Sick Children, Toronto, ON, Canada. [8]Department of Biochemistry, University of Toronto, Toronto, ON, Canada. [9]Division of Gastroenterology, Hepatology and Nutrition, Department of Pediatrics, Hospital for Sick Children, Toronto, ON, Canada. [10]SickKids IBD Centre, Hospital for Sick Children, Toronto, ON, Canada. [11]Department of Medical Biophysics, University of Toronto, Toronto, ON, Canada. ✉e-mail: john.brumell@sickkids.ca

protein (GAP) domain that promotes Guanosine-5'-triphosphate (GTP) hydrolysis by RAB10. By converting RAB10 to a GDP-bound state, SopD promotes recruitment of Dynamin-2 to drive the scission of the PM and also the removal of RAB10 from invasion sites.

SopB, another SPI-1 T3SS effector, was shown to recruit RAB10 to invasion sites at the earliest stages (<10 min) of infection[4]. Thus, we reasoned that the origin of RAB10 may provide insight into the membrane source for ruffle formation (Fig. 1a). Here, we demonstrate that RAB10[+] tubular structures exist prior to infection and serve as membrane reservoirs that are mobilized upon STm infection. These membrane reservoirs are preloaded with certain exocyst subunits and are mobilized in a SopB-dependent manner. Furthermore, we show that SPI-1 effectors SopB, SipC, and SopE2 act cooperatively to recruit exocyst subunits from different cellular compartments. Together,

these independent effector-driven pathways contribute to the formation of invasion ruffles and subsequent uptake of bacteria into host cells.

## Results

### SopB mediates disassembly of pre-existing RAB10[+] tubules and delivery of RAB10 to STm invasion sites

We examined RAB10 localization in a human intestine-derived epithelial cell line (Henle 407). Cells were transfected with GFP-RAB10 and RFP-LifeAct (F-actin probe) and then infected with BFP-expressing STm. In control cells, GFP-RAB10 localized to tubular structures, consistent with prior observations[5,6]. However, during infection, these tubules displayed a trend of disassembly that was concomitant with rapid delivery of RAB10 to STm invasion sites (Fig. 1b and

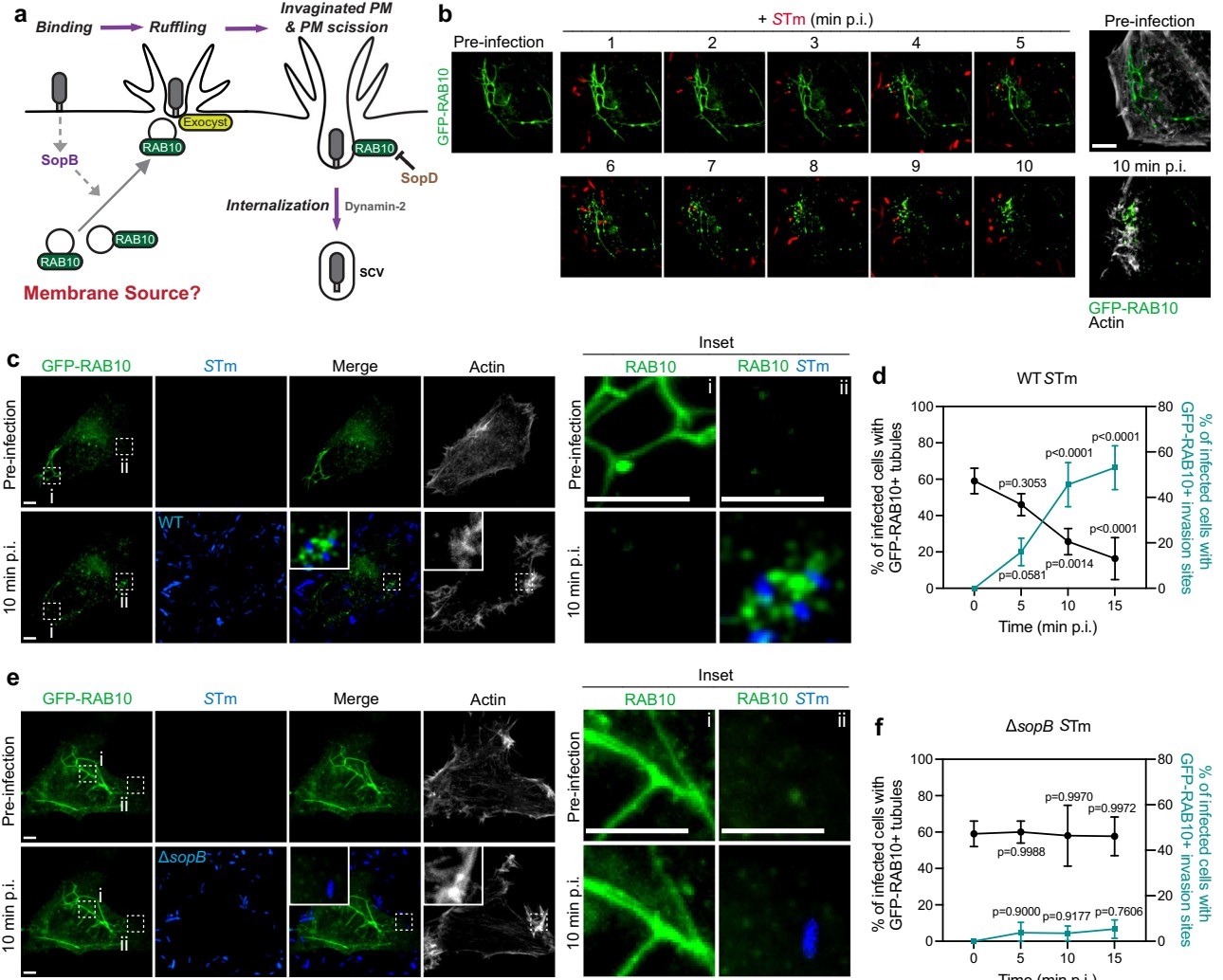

**Fig. 1 | SopB mediates the disassembly of RAB10[+] tubules and delivery of RAB10 to STm invasion sites. a** Current model of SopB-mediated RAB10[+] vesicle recruitment to invasion sites. **b** Representative live images of RAB10[+] tubules in WT Henle 407 cells, pre-infection and at indicated time points post-WT BFP-STm (in red) infection (p.i.). RFP-LifeAct and BFP-STm were used to identify invasion sites in real-time. Data are representative of three independent experiments.
**c, d** Representative images (**c**) and quantifications (**d**) of RAB10[+] tubules and RAB10[+] invasion sites in either pre-infection or WT STm-infected Henle 407 cells for the indicated time. RFP-LifeAct and BFP-STm were used to identify invasion sites in real-time. Inset 'i' indicates an ROI with RAB10[+] tubules while inset 'ii' indicates an ROI of a representative invasion site. n = 3 Independent experiments. At least 100 cells for each condition in each experiment were scored for the presence or

absence of RAB10-positive membrane reservoirs or 100 invasion sites were scored for RAB10 recruitment. **e, f** Representative images (**e**) and quantifications (**f**) of RAB10[+] tubules and RAB10[+] invasion sites in Henle 407 cells pre-infection and at indicated time points post ΔsopB STm infection. RFP-LifeAct and BFP-STm were used to identify invasion sites in real-time. Inset 'i' indicates an ROI with RAB10[+] tubules while inset 'ii' indicates an ROI of a representative invasion site. n = 3 Independent experiments. At least 100 cells for each condition in each experiment were scored for the presence or absence of RAB10-positive membrane reservoirs or 100 invasion sites were scored for RAB10 recruitment. Data shown are means ± standard deviation (SD). P value was calculated using (**d, f**) two-way ANOVA with Tukey's HSD post-hoc test. Scale bars, **b** 8 μm, **c** and **e** 5 μm. Source data are provided as a Source Data file.

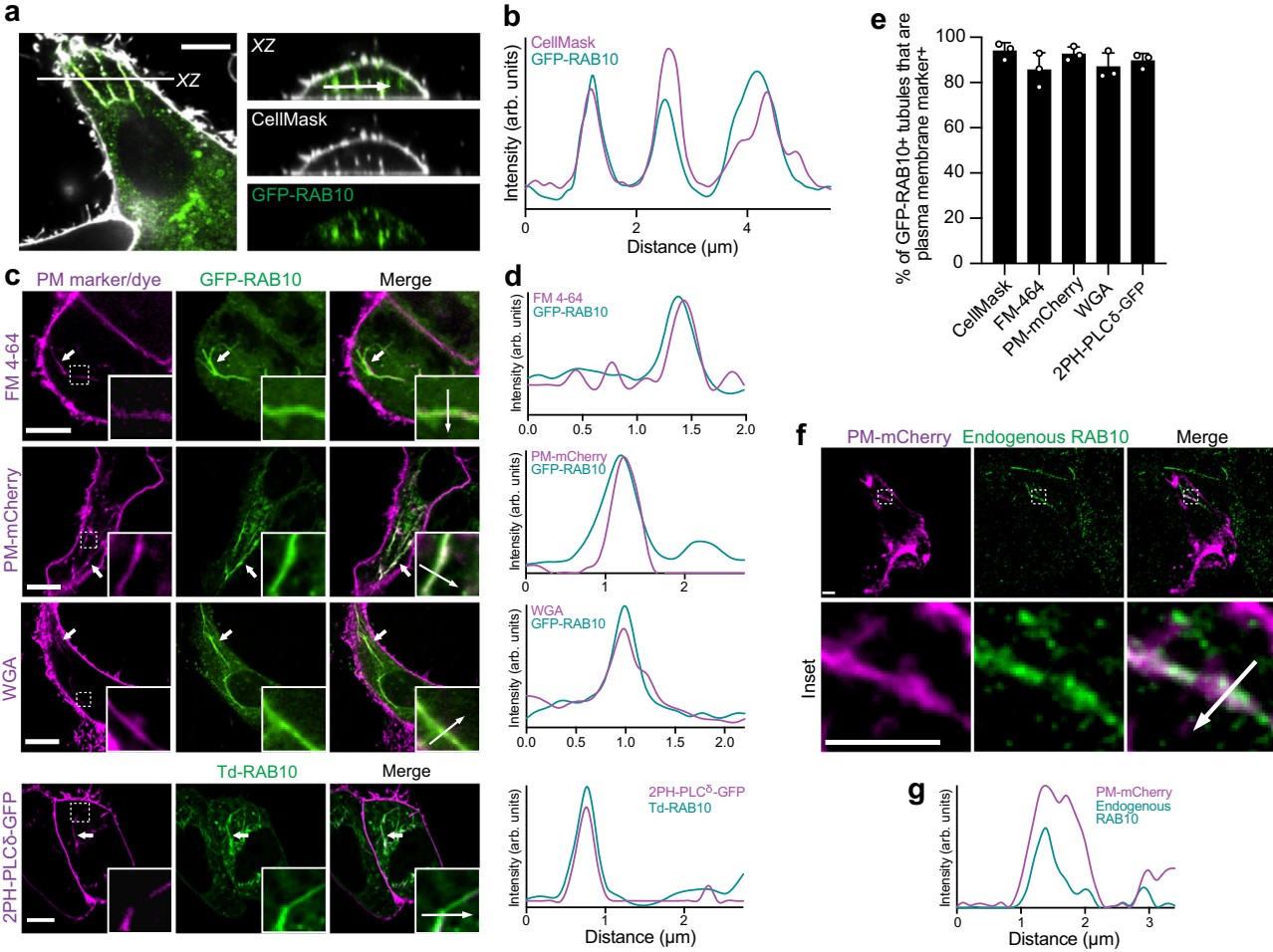

**Fig. 2 | RAB10⁺ tubular membrane reservoirs are PM-associated.**
**a** Representative images of WT Henle 407 cells transfected with GFP-RAB10 and stained for CellMask. *XZ* section was chosen to depict RAB10⁺ tubules' colocalization with CellMask. Mander's correlation coefficient (MCC) values for the relationship between CellMask (channel 1) and GFP-RAB10 (channel 2) were calculated to indicate colocalizations ($M1 = 0.61 \pm 0.12$, $M2 = 0.54 \pm 0.12$, and mean ± SD). **b** Line plot profile of the white arrow in the *XZ* section in (**a**). In this and the following panels, arb. units (arbitrary units) indicate the signal densities along the chosen white arrow. **c** Representative images of RAB10⁺ tubules in Henle 407 cells positive for indicated plasma membrane (PM) markers and identified as deep PM reservoirs. White arrows and boxes indicate RAB10⁺ deep PM invaginations. In this and the following panels, MCC values were calculated to indicate colocalizations between two channels (the left panel as channel 1 and the right panel as channel 2).

For FM 4-64 (channel 1) and GFP-RAB10 (channel 2), $M1 = 0.56 \pm 0.11$ and $M2 = 0.52 \pm 0.07$. For PM-mCherry and GFP-RAB10, $M1 = 0.58 \pm 0.07$ and $M2 = 0.53 \pm 0.08$. For WGA and GFP-RAB10, $M1 = 0.57 \pm 0.07$ and $M2 = 0.53 \pm 0.06$. For 2PH-PLCδ-GFP and Td-RAB10, $M1 = 0.59 \pm 0.05$ and $M2 = 0.51 \pm 0.10$. **d** Line plot profiles of the white arrows in the insets in (**c**). **e** Quantifications of RAB10⁺ tubules in Henle 407 cells were positive for indicated PM markers and identified as deep PM reservoirs. $n = 3$ Independent experiments with 100 cells examined in each experiment. **f** Representative images of WT Henle 407 cells transfected with PM-mCherry and stained for endogenous RAB10. The inset (lower panel) was chosen to depict endogenous RAB10+ tubules' colocalization with PM-mCherry. MCC values for PM-mCherry and endogenous RAB10 are $M1 = 0.52 \pm 0.08$ and $M2 = 0.51 \pm 0.10$. **g** Line plot profile of the white arrow in the inset in (**f**). Data shown are means ± SD. Scale bars, 10 µm. Source data are provided as a Source Data file.

Supplementary Movie 1). Indeed, as early as 10 min post-infection (p.i.), we could observe the complete disassembly of tubules in one region of the cell and recruitment of RAB10 to distal *S*Tm invasion sites (Fig. 1c, d). Cells infected with an *S*Tm mutant lacking SopB (Δ*sopB*, deleted for the *sopB* gene) retained RAB10⁺ tubules and did not display RAB10 recruitment to invasion sites (Fig. 1e, f). These findings suggest that RAB10 is recruited from pre-existing tubules to invasion sites through a mechanism requiring SopB.

### RAB10 and its effectors generate membrane reservoirs before the infection

Prior studies suggested RAB10 localizes to tubular endosomes[5,6]. However, we observed that known endocytic RAB GTPases (RAB4, RAB5, and RAB11) did not colocalize with RAB10⁺ tubules (Supplementary Fig. 1a). Therefore, we considered the possibility that RAB10 localized to tubular invaginations of the PM, previously characterized as membrane reservoirs[7,8]. To test this, cells were labeled with

CellMask to visualize the cell surface. Remarkably, GFP-RAB10⁺ tubules colocalized with CellMask and appeared to be continuous with the cell surface as demonstrated in the *XZ* cross-section (Fig. 2a, b). RAB10⁺ tubules colocalized with other PM markers, including fluorescent probes (FM4-64 and wheat germ agglutinin-Alexa647 (WGA)) and PM marker proteins, including PM-mCherry (targeted to the PM via the Lyn myristoylation and palmitoylation sequences from Lyn tyrosine kinase[9]) and 2PH-PLCδ-GFP (a probe for phosphatidylinositol-(4,5)-bisphosphate[10]) (Fig. 2c, d). In all cases, over 85% of RAB10⁺ tubules colocalized with PM markers (Fig. 2e). By immunofluorescence staining with RAB10 antibodies, we observed colocalization of endogenous RAB10 with tubular membrane reservoirs (PM-mCherry⁺) (Fig. 2f, g) in Henle 407 cells under basal growth conditions. In contrast, endogenous RAB4, RAB5, and RAB11 did not colocalize with membrane reservoirs (Supplementary Fig. 1b, c). We also observed RAB10⁺ tubular invaginated PM compartments (CellMask⁺) in multiple cell types, including human SH-SY5Y, MCF-7, T84, Caco-2, HeLa, HEK293 cells and

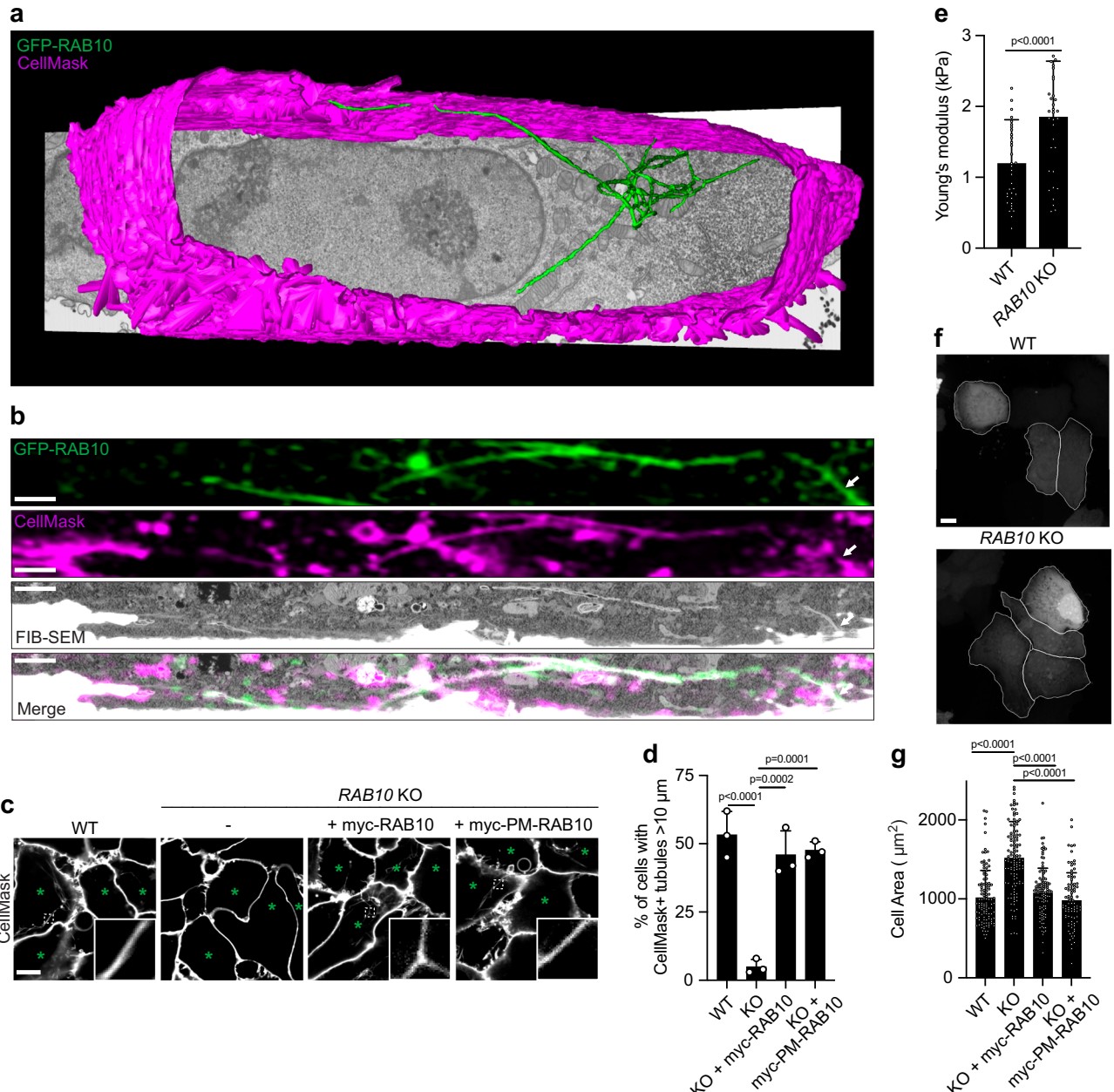

**Fig. 3 | RAB10 acts locally to generate PM-associated membrane reservoirs and regulate PM biophysical properties. a** 3D rendering of RAB10⁺ PM reservoirs and the PM from a large volume FIB-SEM dataset. **b** Representative image of a single slice from fluorescence images correlated to the same cell imaged by FIB-SEM. The full correlated dataset is in Supplementary Movie 2. **c**, **d** Representative images (**c**) and quantifications (**d**) of deep PM reservoirs (as analyzed for the presence of at least one CellMask⁺ tubular structure >10 μm in length) in WT, *RAB10* KO or *RAB10* KO Henle 407 cells that were transfected with indicated plasmid. GFP-control vectors were co-transfected to indicate transfected cells (marked with green asterisks). *n* = 3 Independent experiments with 100 cells examined in each

experiment. **e** Atomic force microscopy measurements of basal membrane tension of WT and *RAB10* KO Henle 407 cells. *n* = 3 Independent experiments with ten cells examined in each experiment. **f**, **g** Representative images (**f**) and quantifications (**g**) of the cell area of WT, *RAB10* KO or *RAB10* KO Henle 407 cells that were transfected with indicated plasmid. GFP-control vectors were co-transfected to indicate transfected cells and used to quantify cell area. *n* = 3 Independent experiments with 25 cells examined in each experiment. Data shown are means ± SD. *P* value was calculated using **d**, **g** one-way analysis of variance (ANOVA) and **e** two-tailed unpaired *t* test. Scale bars, **b** 2 μm and **c**, **f** 10 μm. Source data are provided as a Source Data file.

mouse embryonic fibroblasts (Supplementary Fig. 1d, e). We conclude that the majority of RAB10⁺ tubules in Henle 407 cells represent invaginated PM compartments.

To further characterize the RAB10⁺ tubules, we used focused ion beam-scanning electron microscopy (FIB-SEM) to obtain the 3D volume of a Henle 407 at high resolution. The cell featured an extensive network of RAB10⁺ tubules that contacted the cell exterior and reached deep into the cell (Fig. 3a and Supplementary Movie 2). To prove that these structures are indeed RAB10⁺ tubules connected

to the cell surface, we utilized correlative light and electron microscopy (CLEM) imaging by labeling these structures with GFP-RAB10 and CellMask and then imaging samples with FIB-SEM. After the acquisition of fluorescence images, samples were processed with contrasting agents (malachite green, ruthenium red, and tannic acid) to enhance visualization of the PM. With the correlation of fluorescence and EM signals, we could observe that these tubules were RAB10⁺ and open to the cell exterior (CellMask⁺) (Fig. 3b and Supplementary Movie 3).

Following *RAB10* knockout (Supplementary Fig. 2a, b), a nearly complete loss of CellMask⁺ tubules was observed (Fig. 3c, d). Expression of epitope-tagged WT RAB10 (myc-RAB10) was sufficient to complement the formation of CellMask⁺ tubules in the *RAB10* KO cells (Fig. 3c, d) and the construct localized to these surface-connected tubules (Supplementary Fig. 3a). Similarly, a RAB10 construct artificially targeted to the PM by replacing its C-terminal cysteines with the polybasic C-terminus of K-Ras (myc-PM-RAB10)[11] was sufficient to complement formation of CellMask⁺ tubules in the *RAB10* KO cells (Fig. 3c, d) and localized to these structures (Supplementary Fig. 3a, b). These findings suggest that RAB10 generates tubular compartments associated with the PM.

Proteins that induce PM invaginations have been proposed to buffer membrane tension[12,13]. Using atomic force microscopy, we observed that *RAB10* KO Henle 407 cells had significantly higher membrane tension than control cells (Fig. 3e). Furthermore, *RAB10* KO cells displayed an increase in cell area (Fig. 3f, g). Expression of either myc-RAB10 or myc-PM-RAB10 was sufficient to restore normal cell size in *RAB10* KO cells (Fig. 3f, g). Our studies are consistent with prior studies showing that PM invaginations serve as membrane reservoirs to control cell surface area[7,8]. Therefore, we refer to RAB10⁺ tubular compartments associated with the PM as RAB10⁺ membrane reservoirs to be consistent with prior work[7,8]. Together, our data suggest that RAB10 is required to maintain the biophysical properties of the PM under normal growth conditions.

The switch between GTP and GDP binding states is a major determinant of subcellular distribution and function for RAB family members[14,15]. To determine the role of RAB10's nucleotide-binding state on PM homeostasis, cells were transfected with constitutively active (Q68L) or dominant negative (T23N)[16] GFP-RAB10 constructs and labeled with CellMask. WT and constitutively active GFP-RAB10 constructs displayed colocalization with CellMask on membrane reservoirs and did not affect their formation (Supplementary Fig. 3c, d). In contrast, the dominant negative GFP-RAB10 mutant was localized to the cytosol and blocked the formation of membrane reservoirs. Our findings suggest that PM-localized RAB10 needs to be in the active GTP-bound state to generate RAB10⁺ membrane reservoirs.

To screen for RAB10 interactors that may play a role in the dynamics of membrane reservoirs, the proximity-based screen method BioID[17,18] was performed with WT, Q68L, and T23N RAB10 (Supplementary Data 1). Consistent with our findings above, RAB10 WT BioID identified many high-confidence interactors known to localize to the PM (Supplementary Fig. 4). As expected, many known RAB10 effectors (e.g., EHBP1[4,19] and MICAL-L1[20]) were detected with the RAB10 Q68L protein but not the T23N mutant, confirming the efficacy of our BioID screen (Fig. 4a). These data are consistent with a prior study in WT Henle 407 cells showing that EHBP1 and MICAL-L1 colocalize with RAB10 on tubules[4]. Here, we found that the knockdown of either effector resulted in the loss of RAB10⁺ membrane reservoirs (Fig. 4b), suggesting that these two effectors play a role in the generation and/or stabilization of these structures.

Proteins of the Bin, Amphiphysin, and Rvs (BAR) domain family serve as scaffolding proteins that can induce and/or stabilize membrane curvature and membrane tubulation[21,22]. Seven BAR domain-containing proteins were detected as novel high-confidence proximity interactors of the RAB10 Q68L mutant (Fig. 4a). Thus, we hypothesized that RAB10-interacting BAR domain proteins may contribute to the establishment of RAB10⁺ membrane reservoirs. Notably, PACSIN2 and PACSIN3 localized to tubular structures (Fig. 4c) and colocalized with RAB10 (Fig. 4d, e). Knockdown of either PACSIN protein inhibited the formation of membrane reservoirs (Fig. 4f). Thus, the BAR domain proteins PACSIN2 and PACSIN3, as well as RAB10's known effectors EHBP1 and MICAL-L1, play a role in the formation of RAB10⁺ membrane reservoirs.

## Exocyst subunit EXOC2 mobilizes from membrane reservoirs to *S*Tm invasion sites

The exocyst is a hetero-octameric complex that tethers secretory vesicles at the PM during exocytosis[23]. In mammals, subunits of the exocyst complex are highly dynamic and exist in tetrameric sub-complexes that can associate independently with vesicles and the PM[24]. The tetrameric complexes are also in dynamic equilibrium with subunit monomers and the hetero-octameric tethering complex[24].

During *S*Tm infection, membrane delivery via recruitment of the exocyst complex to invasion sites is known to be required for ruffle formation[2]. Consistent with these previous findings[2], we observed the recruitment of exocyst components EXOC2 (SEC5), EXOC3 (SEC6) (Fig. 5a), and EXOC7 (EXO70) (Supplementary Fig. 5a) to *S*Tm invasion sites. EXOC7 associates with the PM in normal growth conditions prior to *S*Tm infection[25] (Supplementary Fig. 5b) and is recruited to the invasion sites where it binds directly to SipC, a component of the SPI-1 T3SS translocon[2]. This interaction is believed to provide the spatial targeting landmark for other exocyst subunits and subsequent assembly[25–27]. However, the membrane source of these exocyst component-containing vesicles is unclear.

Since exocyst subunits were identified in our RAB10 BioID analysis (Supplementary Data 1), we hypothesized that RAB10⁺ membrane reservoirs might contain exocyst subunits prior to infection. To test this, we examined the localization of EXOC2 and EXOC3 prior to *S*Tm infection. EXOC2 displayed significant localization on tubular structures while EXOC3 mostly localized to the perinuclear region (Supplementary Fig. 5c). By co-staining with PM-targeted RAB10, we found EXOC2 localized to RAB10⁺ membrane reservoirs (Fig. 5b). Knockout of *RAB10* blocked the formation of EXOC2 tubules, a phenotype that was restored with the complemented myc-PM-RAB10 expression (Fig. 5c and Supplementary Fig. 5d). However, knockout of *EXOC2* (Supplementary Fig. 2c, d) did not affect the formation of RAB10⁺ membrane reservoirs (Supplementary Fig. 5e, f). Thus, RAB10 expression is required for the establishment of membrane reservoirs that can recruit EXOC2, but this exocyst component is not required for reservoir assembly/stabilization. In contrast to EXOC2, EXOC3 was not localized to RAB10⁺ membrane reservoirs (Supplementary Fig. 5g) but did colocalize with Golgi marker GM130[28] (Fig. 5b). Our findings are consistent with prior work showing that exocyst subunits can have different localization in basal growth conditions[24].

A previous study[2] and our data above-identified exocyst subunit recruitment to *S*Tm invasion sites. However, our findings suggest the possibility that exocyst subunits may be trafficked to *S*Tm invasion sites separately, instead of as a pre-assembled complex. To test this, we examined the recruitment of EXOC2 or EXOC3 to *S*Tm invasion sites. Compared to WT cells, significantly less EXOC2 recruitment was observed in *RAB10* KO cells (Fig. 5d, e). However, the knockout of *EXOC3* (Supplementary Fig. 2e, f) did not affect EXOC2 recruitment (Fig. 5d, e). These data suggest that EXOC2 redistribution from membrane reservoirs to invasion sites depends on RAB10 expression. In contrast, EXOC3 recruitment to invasion sites was not significantly affected by knockout of *RAB10* or *EXOC2* (Fig. 5f and Supplementary Fig. 6a). Together, our data suggests EXOC2 and EXOC3 are recruited to *S*Tm invasion sites through independent pathways.

## SPI-1 T3SS effectors act cooperatively to recruit the exocyst to *S*Tm invasion sites

To determine how *S*Tm mediates the recruitment of exocyst components to the invasion sites, we infected cells with bacterial strains lacking specific SPI-1 T3SS effectors. Previous studies have shown that SipC directly recruits exocyst components to invasion sites[2]. Consistent with this, the *S*Tm mutant strain lacking SipC (Δ*sipC*) had a significant defect in both EXOC2 and EXOC3 recruitment to invasion sites relative to WT bacteria (Supplementary Fig. 7a–d). SopE2 is also reported to contribute to exocyst recruitment to invasion sites[2]. Here,

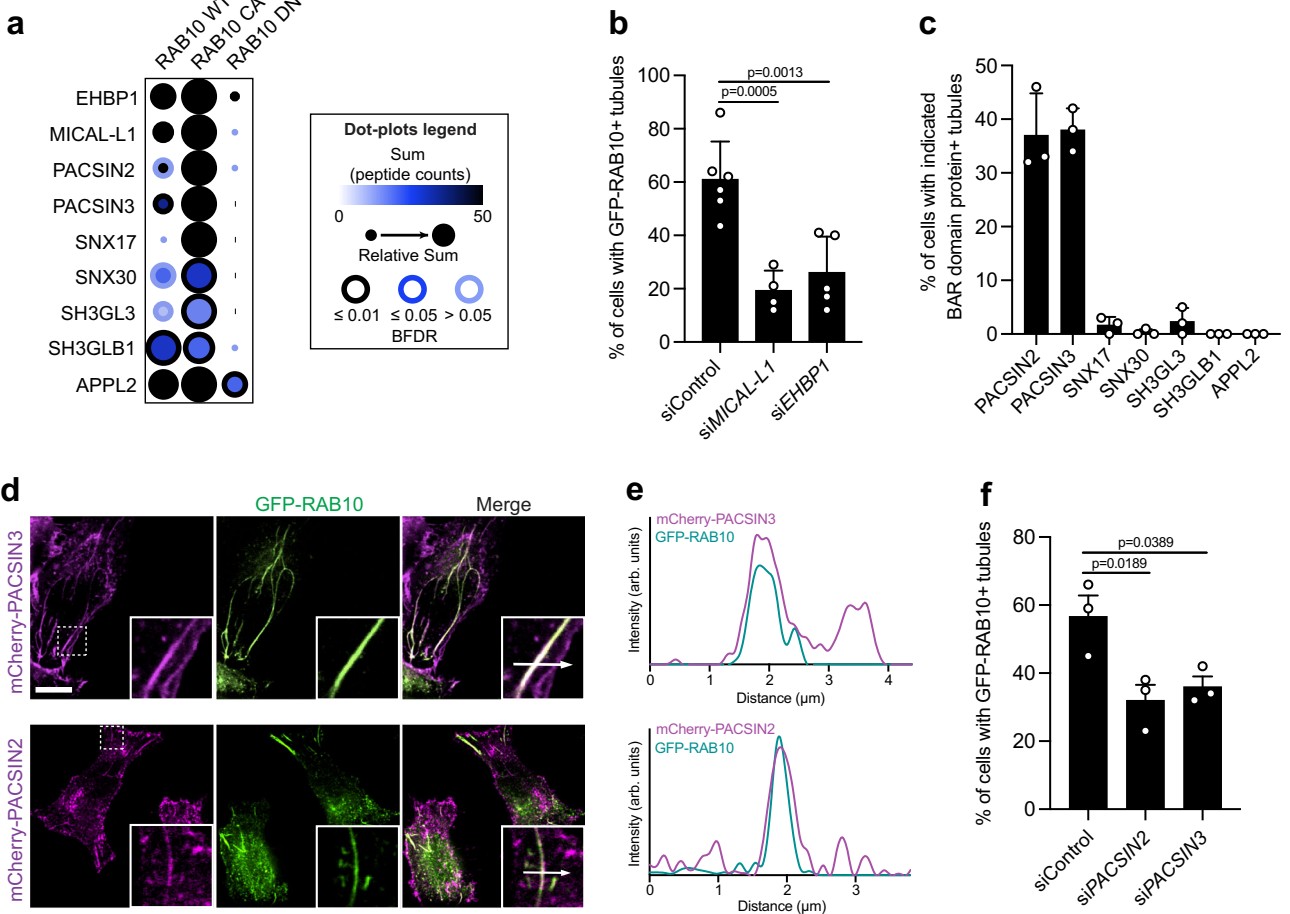

**Fig. 4 | RAB10 effectors contribute to the formation of RAB10⁺ membrane reservoirs. a** The ProHits-viz web tool was used to generate the dot plot view of chosen known and novel RAB10 interactors from the BioID profiling, displaying prey abundance across baits and prey confidence. **b** Quantifications of RAB10⁺ membrane reservoirs in Henle 407 cells with siRNA knockdown of *MICAL-L1* or *EHBP1*. WT Henle 407 cells were transfected with the indicated siRNA and then transfected with GFP-RAB10 24 h later. *n* = 4–6 Independent experiments with at least 100 cells examined in each experiment. **c** Quantifications of BAR domain protein positive tubules in Henle 407 cells transfected with indicated plasmid. *n* = 3 Independent experiments with 100 cells examined in each experiment. **d** Representative images of WT Henle 407 cells co-transfected with GFP-RAB10 and

PACSIN2-mCherry or PACSIN3-mCherry. Insets were chosen to depict the effectors' colocalizations on RAB10⁺ membrane reservoirs. MCC values for mCherry-PACSIN3 and GFP-RAB10 are *M1* = 0.42 ± 0.08 and *M2* = 0.61 ± 0.08. For mCherry-PACSIN2 and GFP-RAB10, *M1* = 0.39 ± 0.09 and *M2* = 0.42 ± 0.11. **e** Line plot profiles of the white arrows in the insets in (**d**). **f** Quantifications of RAB10⁺ membrane reservoirs in Henle 407 cells with siRNA knockdown of *PACSIN2* or *PACSIN3*. WT Henle 407 cells were transfected with the indicated siRNA and then transfected with GFP-RAB10 24 h later. *n* = 3 Independent experiments with 100 cells examined in each experiment. Data shown are means ± SD. *P* value was calculated using (**b**, **f**) one-way ANOVA. Scale bars, 10 μm. Source data are provided as a Source Data file.

we observed that EXOC3 but not EXOC2 recruitment is diminished during infection of Δ*sopE2* mutant bacteria compared to WT *S*Tm (Supplementary Fig. 7a–d). These data suggest that SopE2 activity is required to mobilize the exocyst subunit from the Golgi membrane instead of RAB10⁺ membrane reservoirs. SopB was shown to mobilize intracellular vesicles to invasion sites, including the recruitment of RAB10⁺ vesicles[4]. Here, we found that the recruitment of both EXOC2 and EXOC3 is dependent on SopB expression by bacteria (Supplementary Fig. 7a–d). However, an *S*Tm mutant lacking SopD, an SPI-1 T3SS effector with GAP activity towards RAB10[4], did not significantly affect exocyst subunit recruitment (Supplementary Fig. 7a–d). Together, these data suggest that although exocyst subunits reside in different cellular compartments prior to infection, SPI-1 T3SS effectors act cooperatively to recruit and assemble the exocyst complex at *S*Tm invasion sites.

### SopB recruits EXOC2 and EXOC3 to invasion sites via independent pathways

To further examine how SopB mediates the recruitment of different exocyst components from different cellular compartments, we

infected WT Henle 407 cells with Δ*sopB S*Tm complemented with WT or mutated SopB constructs (Fig. 6a). C460 is essential for SopB's catalytic activity towards phosphatidylinositol polyphosphates and inositol polyphosphates[29,30], which is known to direct the recruitment of RAB10 and other host factors to invasion sites[4,31,32]. Here, we observed that complementation of the Δ*sopB* bacteria with WT SopB, but not the catalytically inactive C460S mutant restored recruitment of both EXOC2 and EXOC3 to invasion sites at 10 min p.i. (Fig. 6b–d and Supplementary Fig. 8a, b).

SopB is known to directly bind to CDC42[33–35] though the importance of this interaction is unclear. We observed that CDC42 colocalized with EXOC3 on the Golgi in control cells and both proteins were recruited to invasion sites within 10 min (Fig. 6e) in a SopB-dependent manner (Supplementary Fig. 9). We mutated L76 and L84 of SopB, which are essential for its binding with CDC42[33,35], and complemented Δ*sopB* bacteria with these two mutants. Under these conditions, L76P or L84P SopB were unable to restore recruitment of CDC42 to invasion sites at 10 min p.i. (Supplementary Fig. 9). L76P and L84P SopB restored recruitment of EXOC2 but not EXOC3 to invasion sites (Fig. 6b–d and Supplementary Fig. 8). This finding suggested that

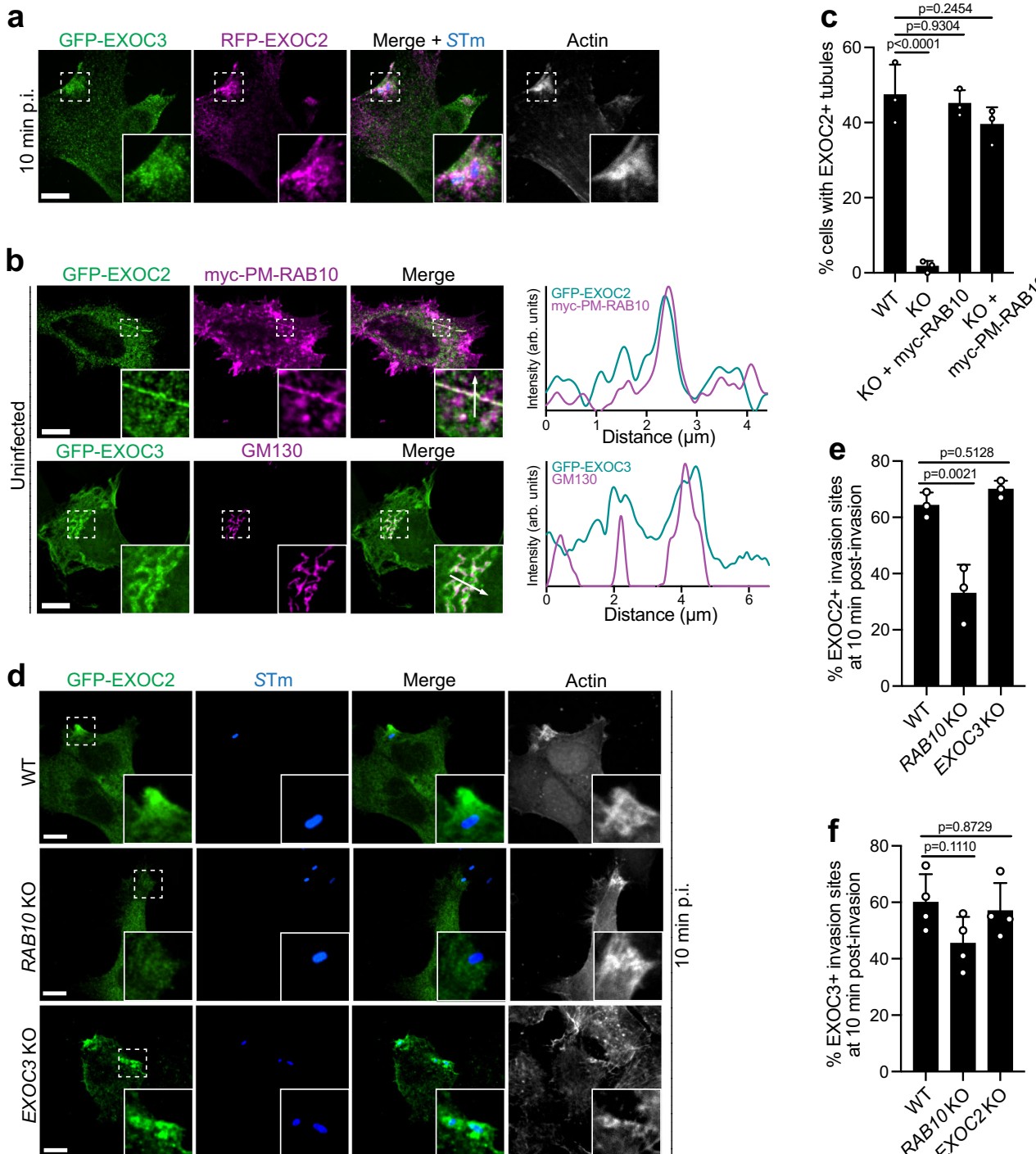

**Fig. 5 | Exocyst subunits EXOC2 and EXOC3 are independently recruited to STm invasion sites. a** Representative images of WT Henle 407 cells transfected with GFP-EXOC3 and RFP-EXOC2 constructs and infected with WT STm. Cells were fixed and imaged at 10 min post-infection. In this and the following panels, invasion sites were identified by actin and STm staining. Data are representative of three independent experiments. **b** WT Henle 407 cells were transfected with GFP-EXOC2 and myc-PM-RAB10 (upper panel) or GFP-EXOC3 (lower panel) and then fixed and stained for myc-tag (upper panel) or GM130 (lower panel). Representative images and line plot profiles (white arrows in insets) depicting the localization of EXOC2 on RAB10+ tubular membrane reservoirs and EXOC3 on Golgi membrane. MCC values for GFP-EXOC2 and myc-PM-RAB10 are $M1 = 0.46 \pm 0.08$ and $M2 = 0.44 \pm 0.07$. For GFP-EXOC3 and GM130, $M1 = 0.24 \pm 0.06$ and $M2 = 0.61 \pm 0.07$. Data are representative of three independent experiments. **c** Quantifications of the percentage of

cells with EXOC2+ tubules, in WT, *RAB10* KO or *RAB10* KO Henle 407 cells with myc-PM-RAB10 overexpression. $n = 3$ Independent experiments with 100 cells examined in each experiment. **d, e** Representative images (**d**) and quantifications (**e**) of EXOC2 recruitment to STm invasion sites. WT, *RAB10* KO or *EXOC3* KO Henle 407 cells were transfected with GFP-EXOC2 and then infected with WT STm. Cells were fixed and imaged at 10 min post-infection. $n = 3$ Independent experiments with 100 invasion sites examined in each experiment. **f** Quantifications of EXOC3 recruitment to STm invasion sites. WT, *RAB10* KO or *EXOC2* KO Henle 407 cells were transfected with GFP-EXOC3 and infected with WT STm. Cells were fixed and imaged at 10 min post-infection. $n = 3$ Independent experiments with 100 invasion sites examined in each experiment. Data shown are means ± SD. *P* value was calculated using one-way ANOVA. Scale bars, 10 μm. Source data are provided as a Source Data file.

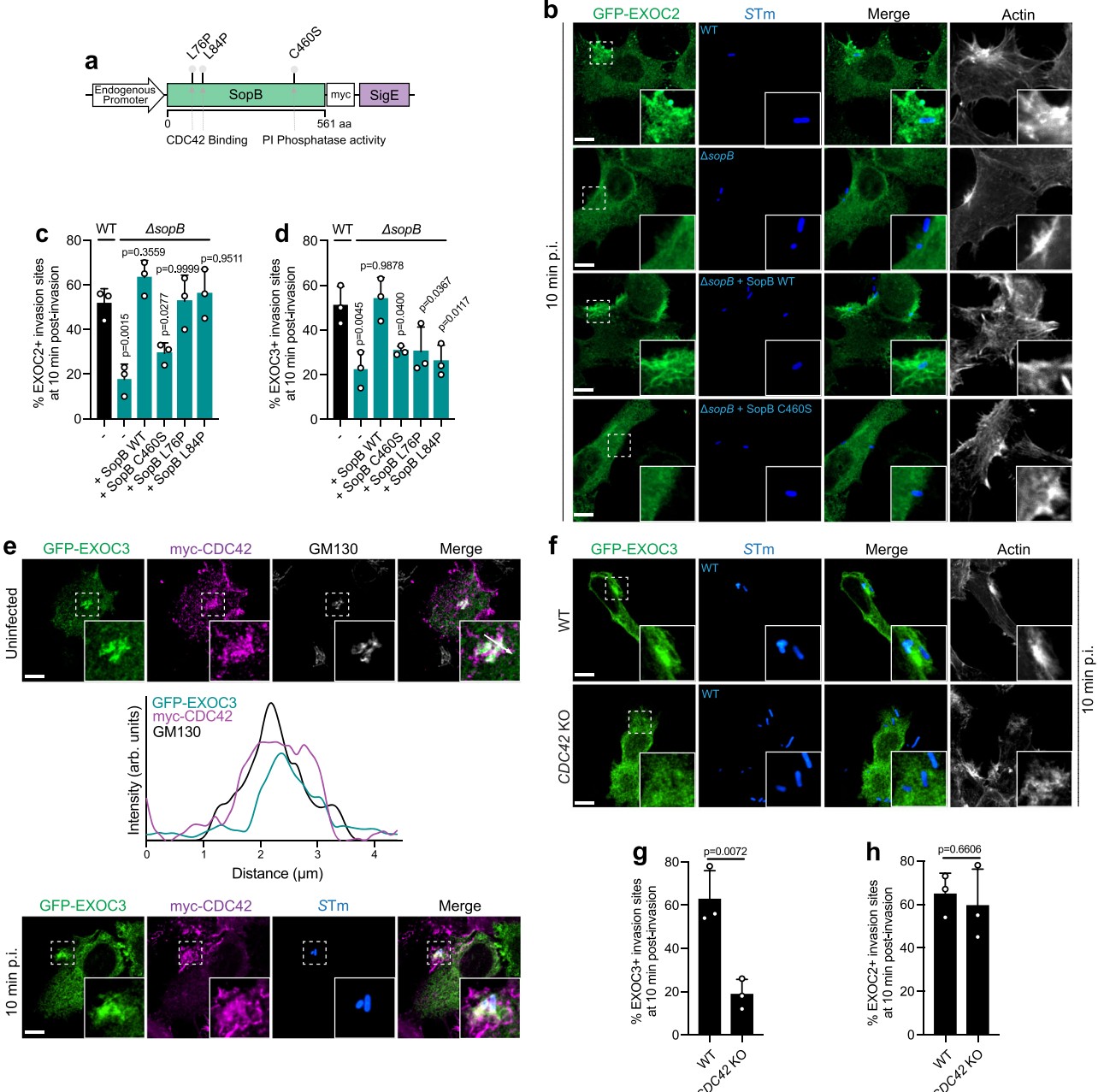

**Fig. 6 | SopB recruits EXOC2 and EXOC3 to STm invasion sites via independent pathways. a** Protein map of SopB depicting the mutation sites introduced in this study: L76P and L84P were mutated to interrupt SopB's direct binding with CDC42 while C460S was mutated to disrupt SopB's PI phosphatase activity.
**b, c** Representative images (**b**) and quantifications (**c**) of EXOC2 recruitment to STm invasion sites. WT Henle 407 cells were transfected with GFP-EXOC2 and then infected with indicated STm strains. Cells were fixed and imaged at 10 min post-infection. *n* = 3 Independent experiments with 100 invasion sites examined in each experiment. **d** Quantifications of EXOC3 recruitment to STm invasion sites. WT Henle 407 cells were transfected with GFP-EXOC3 and then infected with indicated STm strains. Cells were fixed and imaged at 10 min post-infection. *n* = 3 Independent experiments with 100 invasion sites examined in each experiment.
**e** Representative images and line plot profiles (white arrows in insets in upper panel) depicting EXOC3 and CDC42's colocalization on Golgi membrane (GM130⁺) prior to STm infection and recruited to invasion site with STm infection. WT Henle

were transfected with GFP-EXOC3 and myc-CDC42 and stained for GM130 and myc-CDC42 (upper panel), or were infected with WT STm and fixed and imaged at 10 min post-infection, actin and STm staining were used to identify invasion sites (lower panel). MCC values for GFP-EXOC3 and myc-CDC42 are *M1* = 0.55 ± 0.02 and *M2* = 0.42 ± 0.08. Data are representative of three independent experiments.
**f, g** Representative images (**f**) and quantifications (**g**) of EXOC3 recruitment to STm invasion sites. WT or *CDC42* KO Henle 407 cells were transfected with GFP-EXOC3 and then infected with WT STm. Cells were fixed and imaged at 10 min post-infection. *n* = 3 Independent experiments with 100 invasion sites examined in each experiment. **h** Quantifications of EXOC2 recruitment to STm invasion sites. WT or *CDC42* KO Henle 407 cells were transfected with GFP-EXOC2 and then infected with WT STm. Cells were fixed and imaged at 10 min post-infection. *n* = 3 Independent experiments with 100 invasion sites examined in each experiment. Data shown are means ± SD. *P* value was calculated using **c**, **d** one-way ANOVA and **g**, **h** two-tailed unpaired *t* test. Scale bars, 10 μm. Source data are provided as a Source Data file.

SopB-CDC42 binding is required for EXOC3 recruitment to *S*Tm invasion sites. Consistent with this notion, EXOC3 recruitment to invasion sites was impaired in *CDC42* knockout cells (Fig. 6f, g and Supplementary Fig. 2g, h) but EXOC2 recruitment was not affected (Fig. 6h and Supplementary Fig. 10). RAB10 recruitment to invasion sites was not affected by the loss of SopB-CDC42 binding (Supplementary Fig. 11). Together, these data suggest that SopB recruits EXOC2 and EXOC3 to invasion sites via independent pathways: the mobilization of EXOC2 from RAB10[+] membrane reservoirs via SopB's catalytic activity and the mobilization of EXOC3 from the Golgi via both SopB's catalytic activity and through direct binding to CDC42.

### Mobilization of membrane reservoirs promotes *S*Tm invasion

The recruitment of the exocyst complex is known to be functionally important for ruffle formation and bacterial entry, as the knockdown of several exocyst subunits, including EXOC2, has been found to reduce ruffle size and bacterial invasion[2]. Furthermore, CDC42 has been shown to contribute to ruffle formation and invasion during *Listeria monocytogenes* infection[36]. Thus, we wanted to determine if RAB10[+] membrane reservoirs contribute to the formation of invasion ruffles. WT cells and cells lacking expression of RAB10, CDC42, EXOC2, or EXOC3 were infected with WT *S*Tm and invasion ruffles were measured with scanning electron microscopy (Fig. 7a, b). All four knockouts significantly reduced invasion ruffle formation by host cells. Similarly, the live-cell measurements of invasion ruffle volume showed that the knockouts resulted in smaller ruffle formation (Fig. 7c, d). Furthermore, bacterial invasion, as measured by colony-forming units, was decreased in each knockout compared to WT cells (Fig. 7e). Altogether, these data support the idea that RAB10[+] membrane reservoirs are important for invasion ruffle formation.

## Discussion

Our findings suggest a model whereby *S*Tm exploits RAB10[+] membrane reservoirs for the invasion of host cells (Fig. 7f). The nature of these reservoirs is remarkable and expands our understanding of RAB10 function[37]. RAB10[+] tubular structures have been described as tubular endosomes in several previous studies[5,6]. Here, we show that none of the well-characterized endocytic RAB GTPases localize to these RAB10[+] tubules. Using fluorescence-based labeling and correlative light-EM imaging, we show that pre-existing RAB10[+] tubules are (at least intermittently) connected to the PM and accessible to extracellular probes, and yet are a separate compartment primed for rapid fusion by an exocyst-dependent process. Under normal growth conditions, RAB10 acts at the cytosolic face of the PM in its GTP-bound state to induce and stabilize membrane reservoirs, thereby controlling PM tension and cell surface area. Our findings are consistent with prior work revealing that membrane reservoirs contribute to PM homeostasis through the regulation of PM expansion[7].

Prior studies have identified a wide spectrum of RAB10 activities in multiple intracellular compartments, including endosomes, the Golgi/TGN, the endoplasmic reticulum, GLUT4-containing exocytic vesicles, neuronal axons, and primary cilia[5,6,37]. Plus, a recent study from Kawai et al. showed that in macrophages RAB10[+] tubular structures can represent a novel endocytic pathway that diverges from canonical micropinocytosis under certain conditions (e.g., PI3-kinase inhibition)[38]. Here, our findings offer fresh perspectives on the multifaceted functions of RAB10 and membrane reservoirs in the context of bacterial invasion. It will be valuable to explore RAB10's roles in regulating these PM-associated tubular reservoirs and PM dynamics in other physiological contexts.

To characterize how these RAB10[+] membrane reservoirs are generated and stabilized, we performed BioID screens with WT or mutant (Q68L and T23N) RAB10 as bait. By comparing these BioID datasets, we show that along with RAB10's known effectors (EHBP1 and MICAL-L1)[19,20], BAR domain proteins PACSIN2 and PACSIN3 play a role in generating these RAB10[+] membrane reservoirs prior to infection. As a family of proteins that possess the ability to sculpt membrane curvature and tubulations[21,22], the role of BAR domain proteins in host membrane rearrangements upon bacterial infection has been previously examined[39]. For example, *S*Tm recruits SNX18 and SNX9 to the membrane ruffles, and that promotes bacterial entry through the effector SopB[40,41]. In future studies, it will be important to determine if any RAB10-interacting proteins are recruited to *S*Tm invasion sites, along with the mobilization of RAB10[+] membrane reservoirs and the trafficking of RAB10[+] vesicles. It will also be important to examine the role of BAR domain proteins in RAB10-mediated ruffle formation and subsequent bacterial invasion.

Another highlight of this study is that we identify the sophisticated interplays between host GTPases and the exocyst complex, which is known to be exploited by bacterial pathogens during infection[42]. Exocyst subunit recruitment and assembly at *S*Tm invasion sites are known to be required for ruffling events[2], although where these subunits originate from was unclear. Here, we show that RAB10[+] membrane reservoirs contain some, but not all exocyst subunits prior to infection, and different exocyst subunits are trafficked from different cellular compartments to *S*Tm invasion sites separately, instead of as a pre-assembled complex. During infection, SopB mobilizes membrane reservoirs and recruits EXOC2 from RAB10[+] membrane reservoirs to invasion sites via its catalytic activity. In parallel, SopB and SopE2 act cooperatively to recruit EXOC3 from the Golgi along with CDC42, an event that requires SopB's CDC42 binding and catalytic activity. Thus, independent effector-driven pathways contribute to exocyst assembly at *S*Tm invasion sites. To obtain the complete picture of how *S*Tm exploits the host exocyst machinery for its invasion, it will be important to investigate the delivery route of other subunits of the exocyst complex not covered in this study.

The cytoskeletal rearrangement and membrane dynamics including drastic ruffle formation are signatures of bacterial invasion used by bacteria with a 'trigger' mechanism[43]. The membrane ruffles extend around the bacteria and fold over, engulfing the bacteria and then fusing back with the membrane[43]. Here we show that *S*Tm exploits host GTPases, RAB10 and CDC42, and exocyst machinery to contribute to ruffle formation. It remains important to characterize whether the mobilization of membrane reservoirs and the delivery of additional membrane material are contingent upon any cytoskeletal alterations. Furthermore, the mechanisms by which SPI-1 T3SS effectors act cooperatively to guide the transport of membrane vesicles is an important question to be answered in the future. Whether other pathogens that use the 'trigger' strategy for bacterial entry can utilize pre-existing membrane reservoirs and manipulate exocyst activity, in a manner similar to *S*Tm, represents an interesting topic for future research.

Following membrane rearrangements at invasion sites and subsequent bacterial uptake into SCVs, *S*Tm also exploits host machinery for membrane trafficking, in favor of its intracellular growth[44]. For example, *S*Tm induces the formation of infection-associated macropinosomes (IAMs) in the vicinity of the SCV[44]. A prior study suggested that these IAMs serve as membrane sources to fuse with SCV which contributes to SCV growth[44]. In the future, it will be interesting to examine if IAM-dependent membrane delivery represents a mechanism that cooperates with the RAB10[+] membrane reservoirs and exocyst delivery to contribute to early infection events.

Since RAB10 and its effectors can stabilize the PM in membrane reservoirs, its recruitment to invasion sites by SopB would be expected to adversely affect pathogen uptake. Indeed, SopD-deficient bacteria are known to become trapped within tubule-shaped invaginated regions of the PM underlying invasion sites and show a PM scission defect that is dependent on RAB10 expression[7]. However, during infection by wildtype *S*Tm, SopD inhibits RAB10 through its GAP activity. By converting RAB10 to its GDP-bound form, SopD promotes Dynamin-2 recruitment to invasion sites[7] and prevents

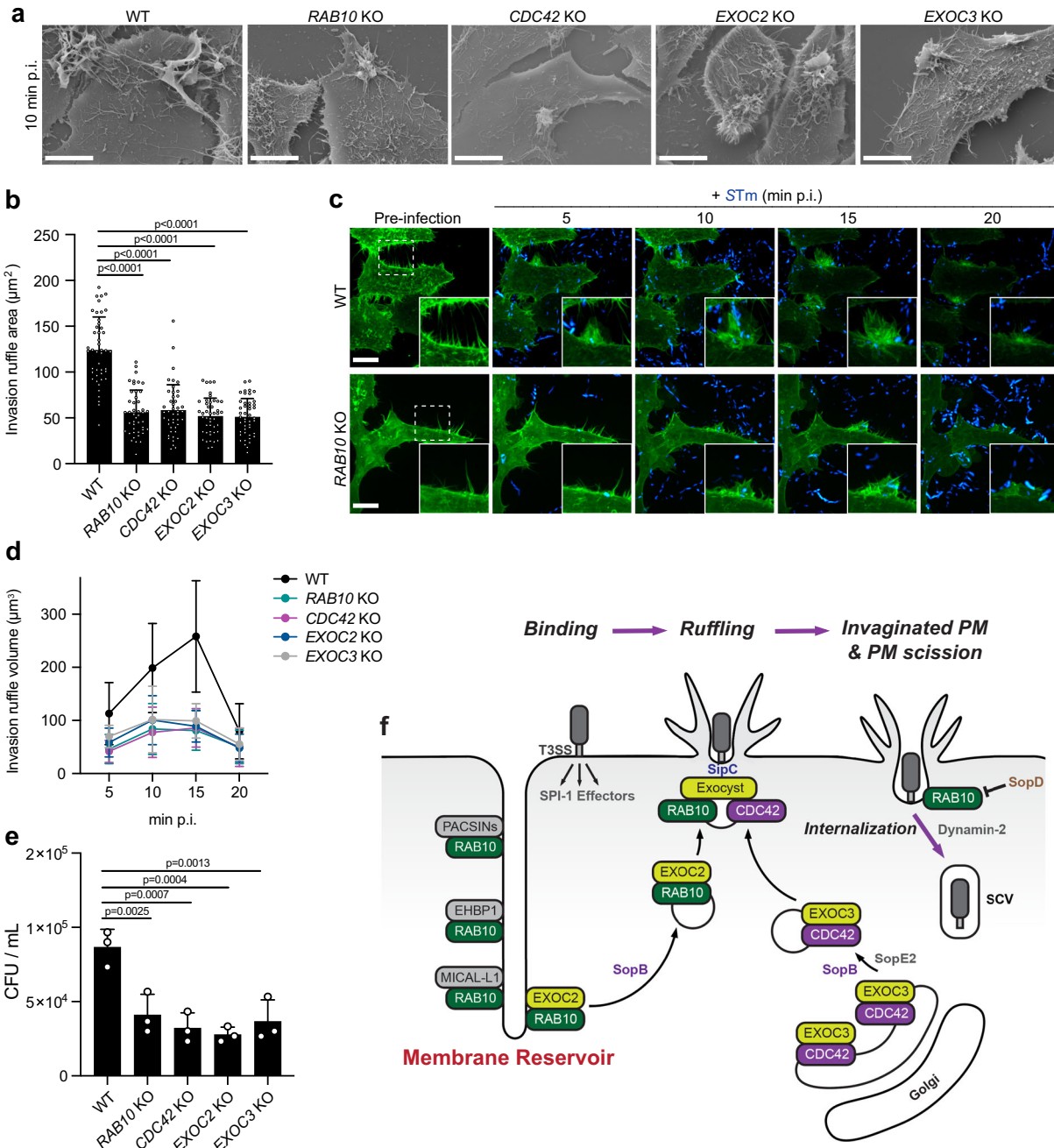

**Fig. 7 | Mobilization of membrane reservoirs promotes STm invasion.**
**a** Representative SEM image of invasion ruffles. WT Henle 407 cells and indicated knockout Henle 407 cells were infected with WT STm and fixed at 10 min post-infection. **b** Quantifications of the area of invasion ruffles identified as in (**a**). The areas of individual invasion ruffle were measured as described in "Methods". $n = 3$ Independent experiments with 15 invasion ruffles examined in each experiment. **c** Representative images of invasion ruffles at indicated time points WT STm post-infection. WT and *RAB10* KO Henle 407 cells were transfected with GFP-LifeAct and infected with WT BFP-STm. GFP-LifeAct and BFP-STm were used to identify invasion sites. **d** Quantifications of the volume of invasion ruffles are identified as in (**c**). The volumes of individual invasion ruffles were measured as described in "Methods". $n = 3$ Independent experiments with ten invasion ruffles examined in each experiment. **e** WT Henle 407 cells and indicated knockout Henle 407 cells were infected with WT STm and lysed at 2 h post-infection for CFU counting, $n = 3$. **f** Model depicting how STm exploits RAB10$^+$ membrane reservoirs for the invasion of host cells. Data shown are means ± SD. *P* value was calculated using (**b**, **e**) one-way ANOVA. Scale bars, 10 μm. Source data are provided as a Source Data file.

membrane reservoir formation. SopB also contributes to PM scission at invasion sites via dephosphorylation of phosphatidylinositol-(4,5)-bisphosphate[45]. Therefore, the combined activities of SopB, SopE2, and SipC on promoting membrane mobilization and the promotion of scission by SopB and SopD allow STm to efficiently invade host cells. Our findings provide new insight into the mechanisms regulating PM homeostasis in mammalian cells and how the cooperative actions of SPI-1 T3SS effectors subvert membrane trafficking during infection.

## Methods

### Plasmids

WT pKH3-tdTomato-RAB10 (Td-RAB10) was obtained from Dr. Zhen-Ge Luo[46]. WT, CA (Q68L) and DN (T23N) peGFP-RAB10 were obtained from Dr. Marci Scidmore[47]. The myc-RAB10 and myc-PM-RAB10 constructs were obtained from Dr. Suzanne Pfeffer[11]. The GFP-RAB4 and GFP-RAB11 constructs were obtained from Dr. Marci Scidmore. The GFP-RAB5 construct was obtained from Dr. Craig Roy. For pCMV-PM-

mCherry, myristoylation and palmitoylation sequences from Lyn tyrosine kinase[9] were fused to mCherry. The restriction sites BamHI and NotI in the peGFP-N1 vector were used for cloning (with the eGFP removed) using the primer pair, 5′-TGCAGGATCCGCCACCATGGGCT GCATTAAAAGCAAACGCAAAGATATGGTGAGCAAGGGCGAGGAGGAT AACATG-3′ and 5′-TGCAGCGGCCGCTTACTTGTACAGCTCGTCCAT GCCGCCGGT-3′. The pEGFP-C1 vector was obtained from Clontech. The mCherry vector was obtained from Dr. Matthew Welch[48]. The 2PH-PLCδ-GFP construct was obtained as a gift from Dr. Mario J. Rebecchi[10]. For BioID, RAB10 (WT, constitutively active (Q68L) and dominant negative (T23N)) was cloned into the pcDNA5-FRT/TO FlagBirA* vector with an N-terminal tag using AscI/NotI restriction sites added to the RAB10 sequences by PCR with 5′-TATAGGCGCGCCAATGGCGAAGAAG ACGTACGACCTGC-3′ forward and 5′-TTAAGCGGCCGCATCAGCAGC ATTTGCTCTTCCAGCC-3′ reverse primers. pOG44 (Invitrogen) was used in combination with the pcDNA5 FRT/TO vectors for stable integration into the Flp-In cells genome. GFP-EHBP1 was obtained as a gift from Dr. Mark McNiven[19]. GFP-MICAL-L1 was received as a gift from Dr. Steve Caplan[20]. PACSIN2-mCherry and PACSIN3-mCherry were obtained as a gift from Dr. Shiro Suetsugu[49]. mCherry-SNX17, mCherry-SH3GL3, mCherry-SH3GLB1, and mCherry-APPL2 were purchased from SPAPC BioCentre and all constructs were verified by DNA sequencing (TCAG, Toronto, Canada). The WT, D288A and H153D HA-PPM1H constructs were obtained from Dr. Dario Alessi[50]. pEGFP-C3-Sec6 was a gift from Channing Der (Addgene plasmid #53757; http://n2t.net/addgene:53757; RRID:Addgene_53757) and was previously described[51]. pEGFP-C3-Sec5 was a gift from Channing Der (Addgene plasmid #53756; http://n2t.net/addgene:53756; RRID:Addgene_53756) and was previously described[51]. mCherry-EXOC2 was constructed via Gibson assembly using GFP-EXOC2 plasmid as the backbone and swapped out the GFP for mCherry-2. All constructs were verified by DNA sequencing (TCAG).

## Cell culture

Henle 407 cells (ATCC, CCL-6) were obtained from the American Type Culture Collection (ATCC). Although Henle 407 cell cultures have been shown to contain HeLa cells, our Henle 407 cells were used between passages 5 and 25, and maintained a distinct morphology relative to HeLa cells. STR profiling of Henle 407 cells was done with GenePrint 10 System (Promega), by TCAG (Supplementary Table 1). MCF-7 cells (ATCC, HTB-22), Caco-2 cells (ATCC, HTB-37), T84 cells (ATCC, CCL-248), HeLa cells (ATCC, CCL-2), SH-SY5Y cells (ATCC, CRL-2266), HEK293 cells (ATCC, CRL-1573), and MEF cells (ATCC, SCRC-1008) were also obtained from ATCC. Flp-In™ T-REx™ 293 cells (Invitrogen) and the Flp-In system were used for the BioID experiments. *RAB10* KO Henle 407 cells were previously described[4]. *EXOC2* KO and *EXOC3* KO Henle 407 cells were made using the CRISPR/Cas9 system and the details are included in the following "CRISPR knockout" section. *CDC42* KO Henle 407 cells were used and previously described[52]. All cells used were authenticated and tested negative for mycoplasma by ATCC and The Hospital for Sick Children Biobank. Cell cultures were maintained in a growth medium (DMEM with 4.5 g/l glucose (Wisent) supplemented with 10% *v/v* FBS (Wisent) at 37 °C with 5% $CO_2$. For microscopy experiments, Henle 407 cells were seeded in 24-well tissue culture plates containing 12 mm coverslips at a density of $6 \times 10^4$ or $3 \times 10^4$ cells/well at either 24 h or 48 h, respectively, before use. For live cell imaging, cells were seeded in μ-Slide 8-well polymer bottom chambers (ibidi) 24 h before use at a density of $4.0 \times 10^4$ cells/well. For immunoblotting, cells were seeded in 6-well tissue culture plates containing a density of $10 \times 10^4$ cells/well 24 h before any treatment and cell lysate collection.

## Transfections and RNA interference

Transfections were performed using X-treme GENE9 (Roche) or GeneeJuice (Millipore) according to the manufacturer's instructions. For

siRNA-mediated knockdown, cells were seeded in 24-well tissue culture plates at a concentration of $3 \times 10^4$ cells/well 24 h before use. Cells were then transfected with 100 nM siRNA using Lipofectamine RNAiMax (Invitrogen) for 48 h as recommended by the manufacturer. LRRK2-directed siRNAs (#1: SASI_Hs01_00242428, 5′-CAUUAGACCUAC GAAUAAA[dT][dT]-3′, and #2: SASI_Hs01_00242433, 5′-CAUUAGACCU ACGAAUAAA[dT][dT]-3′), EHBP1-directed siRNA (SASI_Hs01_00097428, 5′-GAGAUUGUUCAGCAGGUUA[dT][dT]-3′) and MICAL-L1-directed siRNA (SASI_Hs02_00361239, 5′-GUUUCUGGGAGGCUGGCAA[dT][dT]-3′) were purchased from Sigma. For control knockdown, MISSION® siRNA Universal Negative Control (Sigma-Aldrich, #SIC001) was used.

## CRISPR knockout

To disrupt specific gene expression in Henle 407 cells, specific single-guide RNA (sgRNA) was designed using the online tool http://guides.sanjanalab.org/. Custom sgRNA oligonucleotides were synthesized by Sigma-Aldrich. For EXOC2, the sgRNA sequences used are: #1, 5′-CA CCGCTGTGCTTTGAGGGACACTG-3′ and 5′-AAACCAGTGTCCCTCAAA GCACAGC-3′; #2, 5′-CACCGCCAACAACCTAAAACACAGG-3′ and 5′-AA ACCCTGTGTTTTAGGTTGTTGGC-3′; #3, 5′-CACCGATCACCTTGGATA GTACCCG-3′ and 5′-AAACCGGGTACTATCCAAGGTGATC-3′. For EXOC3, the sgRNA sequences used are: #1, 5′-TGGATACTTGCTGAC CAGAG-3′ and 5′-CTCTGGTCAGCAAGTATCCA-3′; #2, 5′-TACTGAGATG ATGAGGAACG-3′ and 5′-CGTTCCTCATCATCTCAGTA-3′; #3, 5′-ACTTCCTTATAATTTCCAGG-3′ and 5′-CCTGGAAATTATAAGGAAGT-3′. The CRISPR/Cas9 vector pSpCas9 (BB)-2A-Puro (pX459) was obtained from Dr. Chi-Chung Hui[53]. For ligation into the BbsI site of pX459, a CACCG sequence was added to the 5′ flanking sequences of the sense oligonucleotides, and an AAAC sequence was added to the 5′ flanking sequences of the antisense oligonucleotides. The sgRNAs were inserted into the pX459 vector. After DNA sequencing (TCAG), the ligated vectors containing the EXOC2 #2 or EXOC3 #3 sgRNA sequence were selected for the experiments. WT Henle 407 cells were then transfected with the ligated vector and 48 h later the transfected cells were selected by puromycin (2 μg/ml) for another 48 h. Single cells were then transferred into a 96-well plate and allowed to grow until confluent. Knockout efficiency was determined by western blot analysis. Additionally, the types of genomic alteration were determined by DNA sequencing (TCAG) of knockout cells.

## Western blots

Cell lysates were resolved by 10% SDS-PAGE, transferred to PVDF membrane (Bio-Rad), and probed with antigen-specific primary antibodies. The following primary antibodies were used for western blot detection: mouse monoclonal anti-RAB10 (Sigma, SAB5300028, lot PM1009301) at 1:1000, mouse monoclonal (6C5) anti-GAPDH (Millipore, MAB374, lot 3768063) at 1:10,000, rabbit monoclonal anti-EXOC2 (Abcam, ab140620) at 1:1000, mouse monoclonal anti-beta tubulin (Sigma, T4026, lot 128M4790V) at 1:10,000, rabbit polyclonal anti-EXOC3 (Proteintech, 14703-1-AP) at 1:1000, and rabbit polyclonal anti-CDC42 (Cell Signaling, 2462, lot 4) at 1:1000. Blocking was performed with 5% skim milk. For all analyses, HRP-conjugated secondary antibodies were used: peroxidase-conjugated goat anti-rabbit IgG (Jackson ImmunoResearch, 11-035-144, lots 152081 and 163676) or peroxidase-conjugated goat anti-mouse IgG (Jackson ImmunoResearch, 111-035-146, lot 157140). Detection was performed using SuperSignal West Femto Maximum Sensitivity Substrate (Thermo). The results were analyzed using Image Lab v6.1 by BioRad.

## Bacterial strains and infections

Unless indicated, infections were performed with WT *S*Tm SL1344[54] and isogenic mutants lacking the effector of interest. The *S*Tm SL1344 mutant lacking SopB (Δ*sopB*)[32] was described previously. For testing SPI-1 T3SS effectors' involvement in the mobilization of membrane

reservoirs and delivery of vesicles, infections were performed with the WT STm 14028S strain and isogenic mutants lacking the effector of interest. WT and the following STm 14028S mutant strains were received from Dr. Helene Andrews-Polymenis and were previously described[55]. Mutants in the STm 14028S background lacking SopB (ΔsopB), SopD (ΔsopD), SopE2 (ΔsopE2), and SipC (ΔsipC) were generated by Lamda Red recombination and PCR verified[55]. For testing the mechanisms of SopB-mediated mobilization of membrane reservoirs and delivery of vesicles, mutations were generated in sopB encoded within pACYC184 and expressed in a ΔsopB[32] STm SL1344 background, including ΔsopB + pSopB WT, ΔsopB + pSopB L76P, ΔsopB + pSopB L84P, and ΔsopB + pSopB C460S. For live-cell imaging, the WT BFP-STm SL1344 was previously described. Isogenic ΔsopB BFP-STm SL1344 was constructed by transforming ΔsopB strain with BFP-pFPV25.1, a plasmid expressing BFP under the control of the rpsM promoter[4].

A previously established approach was used for infection of epithelial cells, using late-log STm cultures as inocula[56]. Briefly, bacteria were pelleted at 10,000× g for 2 min and resuspended in PBS, pH 7.2. For immunofluorescence staining, the bacteria were diluted 1:50 in PBS and added to cells at 37 °C for 10 min. Cells were then fixed with 2.5% paraformaldehyde (PFA) in PBS at 37 °C for 10 min followed by immunofluorescence staining protocols. For live-cell imaging, fluorescence-tagged bacteria were diluted 1:25 in PBS and added to cells on a live-imaging station for the indicated time. For the CFU replication assay, the bacteria were diluted 1:100 and added to cells at 37 °C for 10 min. Cells were then washed with 3× with PBS and growth medium was added to each well until 30 min at 37 °C. Then, the media was changed to a growth medium containing 100 μg/ml Gentamicin until 2 h p.i. Bacteria were solubilized for the replication assay as follows: (1) cells were washed 2× with PBS; (2) 1 ml 1% TX100, 0.1% SDS in PBS was added to each well; (3) the cells were pipetted up and down 5× using 1 ml Gilson tip; (4) the resuspended cells were transferred to sterile Eppendorf tubes and serially diluted; (5) LB plates were divided into quadrants and the dilution factor and culture plate number was marked; (6) 3–4 drops of 10 μl serially diluted (dilutions: $10^{-4}, 10^{-3}, 10^{-2}$, and $10^{-1}$) cells were deposited into each quadrant and cultured the plate at 37 °C for overnight, and; (7) CFUs were counted the next day.

## BioID

BioID[17] was performed as described previously[18]. Briefly, stable T-REx cell line populations were created using pcDNA5-FlagBirA*-FRT/TO constructs and the accessory plasmid pOG44 (Invitrogen) according to the manufacturer's instructions. The selection was performed using hygromycin B at a concentration of 200 μg/ml. For induction of the gene of interest, tetracycline was used at a concentration of 1 μg/ml, and media was supplemented with biotin (50 μM). Biotinylation was performed for 20 h.

## Biotin–streptavidin affinity purification for mass spectrometry

Cell pellets were resuspended in 10 ml of lysis buffer (50 mM Tris-HCl pH 7.5, 150 mM NaCl, 1 mM EDTA, 1 mM EGTA, 1% v/v Triton X-100, 0.1% w/v SDS, 1:500 protease inhibitor cocktail (Sigma), 1:1000 turbonuclease (Sigma), incubated on an end-over-end rotator at 4 °C for 1 h, sonicated to disrupt any visible aggregates, then centrifuged at 45,000x g for 30 min at 4 °C. The cleared supernatant was transferred to a fresh 15 ml conical tube, 30 μl of packed, pre-equilibrated streptavidin-sepharose beads (GE) were added, and the mixture incubated for 3 h at 4 °C with end-over-end rotation. Beads were pelleted by centrifugation at 2000x g for 2 min and transferred with 1 ml of lysis buffer to a fresh Eppendorf tube. Beads were washed once with 1 ml lysis buffer and twice with 1 ml of 50 mM ammonium bicarbonate, pH 8.3. Beads were transferred in ammonium bicarbonate to a fresh centrifuge tube and washed two more times with 1 ml ammonium bicarbonate. Tryptic digestion was performed by incubating the beads with 1 μg MS grade TPCK trypsin (Promega) dissolved in 200 μl of

50 mM ammonium bicarbonate overnight at 37 °C. The following morning, an additional 0.5 μg trypsin was added, and the beads were incubated for 2 h at 37 °C. Beads were pelleted by centrifugation at 2000x g for 2 min, and the supernatant was transferred to a fresh Eppendorf tube. Beads were washed twice with 150 μl of 50 mM ammonium bicarbonate and wash pooled with the eluate. The sample was lyophilized and resuspended in buffer A (0.1% v/v formic acid). One-fifth of the sample was analyzed per MS run.

## Mass spectrometry

HPLC was conducted using a 2 cm pre-column (Acclaim PepMap 50 mm × 100 μm inner diameter), and a 50 cm analytical column (Acclaim PepMap, 500 mm × 75 μm diameter; C18; 2 μm; 100 Å, Thermo), running a 120 min reversed-phase buffer gradient at 225 nl/min on a Proxeon EASY-nLC 1000 pump in-line with a Thermo Q-Exactive HF quadrupole-Orbitrap mass spectrometer. A parent ion scan was performed using a resolving power of 60,000, and then up to the twenty most intense peaks were selected for MS/MS (minimum ion count of 1000 for activation) using higher energy collision-induced dissociation fragmentation. Dynamic exclusion was activated such that MS/MS of the same $m/z$ (within a range of 10 ppm; exclusion list size = 500) detected twice within 5 s were excluded from analysis for 15 s. For protein identification, Thermo RAW files were converted to the mzXML format using Proteowizard[57] and then searched using X! Tandem[58] and COMET[59] against the human RefSeq Version 45 database (containing 36,113 entries). Data were analyzed using the trans-proteomic pipeline[60,61] and the ProHits software suite (v3.3)[62]. Search parameters specified a parent ion mass tolerance of 10 ppm, and an MS/MS fragment ion tolerance of 0.4 Da, with up to two missed cleavages allowed for trypsin. Variable modifications of +16@M and W, +32@M and W, +42@N-terminus, and +1@N and Q were allowed. Proteins identified with an iProphet cut-off of 0.9 (corresponding to ≤1% FDR) and at least two unique peptides were analyzed with SAINT Express v.3.3.1. Twelve control runs (from cells expressing the Flag-BirA* epitope tag) were collapsed to the two highest spectral counts for each prey and compared to the two biological and two technical replicates of RAB10 WT, Q68L, T23N BioID. High confidence interactors were defined as those with Bayesian false discovery rate (BFDR) ≤ 0.01.

## Immunofluorescence

For all fixed microscopy-based experiments, cells were fixed with 2.5% v/v PFA (EM Sciences) in PBS for 10 min at 37 °C, unless indicated otherwise. Immunostaining was performed as previously described[63] using the following primary antibodies: mouse monoclonal anti-RAB10 (Sigma, SAB5300028, lot PM1009301) at a dilution of 1:100, mouse monoclonal anti-C-myc 9E10 (Thermo, MA1-980, lot XJ358688) at 1:500, mouse monoclonal anti-RAB4 (BD Biosciences, 610888, lot 606-259-1550) at a dilution of 1:100, rabbit polyclonal anti-RAB5 (Santa Cruz, sc-598, lot D0207) at a dilution of 1:100, rabbit polyclonal anti-RAB11 (Invitrogen, 71-5300, lot UD281527) at a dilution of 1:100, mouse monoclonal anti-EXOC7 (KeraFast, ED2001, lot 043019) at a dilution of 1:100, mouse monoclonal anti-GM130 (BD Biosciences, 610822, lot 8054546) at a dilution of 1:100, rabbit polyclonal anti-Salmonella (BD Transduction, 229481, lot 4017189) at a dilution of 1:100, chicken polyclonal anti-GFP (Rockland, 600-901-215S, lot 48932) at a dilution of 1:500, and rabbit polyclonal anti-RFP (Abcam, ab28664, lot 629768) at a dilution of 1:500. The following secondary antibodies were used in this study: Alexa Fluor 488-conjugated goat anti-mouse IgG (Invitrogen, A-11029, lot 2179204), Alexa Fluor 488-conjugated goat anti-rabbit IgG (Invitrogen, A-11034, lot 2541675), Alexa Fluor 488-conjugated goat anti-chicken IgG (Invitrogen, A-32931, lot XB343360), Alexa Fluor 568-conjugated goat anti-mouse IgG (Invitrogen, A-11031, lot 2026148), Alexa Fluor 568-conjugated goat anti-rabbit IgG (Invitrogen, A-11011, lot 2379475), Alexa Fluor 647-conjugated goat

anti-mouse IgG (Invitrogen, A-32728, lot XE344349), and Alexa Fluor 647-conjugated goat anti-rabbit IgG (Invitrogen, A-32733, lot TL272452). The secondary antibodies were used at a dilution of 1:500. CellMask Deep Red PM stain (Thermo) was used as a membrane impermeant dye to identify compartments open to the extracellular space and was used without cell permeabilization. Alexa Fluor 647-labeled wheat germ agglutinin (WGA-647) (Invitrogen) was used as a membrane impermeant dye to identify compartments open to the extracellular space and was used without cell permeabilization. FM 4-64 (Invitrogen) was also used as a membrane impermeant dye to identify compartments open to the extracellular space in live cells. The PM dyes were used according to their corresponding manufacturer's instructions.

## Confocal microscopy

Unless otherwise indicated, cells were imaged using a Quorum spinning disk microscope with a 10× or 20×, 1.0 NA objectives, or a 63×, 1.4 NA oil immersion objective (Leica DMI6000B inverted fluorescence microscope with a Yokogawa spinning disk head and Hamamatsu ORCA Flash4 sCMOS camera) and Volocity 6.3 acquisition software (Quorum). Confocal z-stacks of 0.3 μm were acquired. Images were analyzed with the Volocity software or Fiji v2.14.0 (ImageJ) and then imported and assembled in Adobe Illustrator v25.3.1 for labeling. For live cell imaging, cells were seeded in μ-Slide 8-well glass bottom chambers (ibidi). Twenty-four hours after seeding, growth media was replaced with live cell imaging media (RPMI with L-Glutamine and 25 mM HEPES (Wisent) supplemented with 10% FBS (Wisent)) containing the respective treatment condition. Cells were imaged at 37 °C using a Leica DMI 6000B inverted fluorescence microscope with a Yokogawa spinning disk head and Hamamatsu ImagEM X2 camera. Images were taken with a z-spacing of 0.5 μm. For invasion ruffle volume measurements, confocal z-stacks of 0.3 μm were acquired.

## Confocal images analysis

The images acquired from confocal microscopy were then imported into Fiji v2.14.0 (ImageJ) (NIH)[64].

For the percentage of cells containing at least one RAB10+ tubular membrane reservoir >10 μm in length (tubule length was measured by Fiji), cells were scored manually for the presence or absence of at least one Rab10+ tubule over 10 μm and over 100 cells were analyzed in each condition and repeated for three independent experiments. For colocalization assays, the Mander's correlation coefficient values were calculated for more than four images from each experiment by using Coloc 2, a Fiji plugin for colocalization analysis (http://imagej.net/Coloc_2). For RAB10's colocalization with PM markers on tubular membrane reservoirs, ROIs of single-plane images were manually drawn to exclude the cell border. This was done to eliminate interference from the strong PM marker signal present on the cell border, as it was not the primary focus of the colocalization assay on tubular membrane reservoirs. For other colocalization analyses, whole cell ROIs on maximum intensity z-projected images were chosen to represent the colocalization on the whole cell level. The line plot profiles were obtained by using the plot profile function in Fiji, as previously described[5]. For the STm invasion site recruitment experiment, F-actin and STm staining (or fluorescence-tagged STm) were used to denote the site of invasion at 10 min p.i. An enrichment of the indicated protein's signal at the STm invasion site, relative to the signal in the cytosol, was considered a positive recruitment. For cell area measurements, details are specifically described in the following "Cell area measurements" section. For invasion ruffle volume measurements, ROIs were drawn around individual invasion ruffles (containing one or more bacteria) volume measurements. At least ten invasion ruffles were counted in each condition and repeated for three independent experiments.

## CLEM

WT Henle 407 cells were seeded in 35 mm ibidi μ-dishes with the Grid-50 location grid (ibidi). After 24 h the cells were transfected with GFP-RAB10 as described above. After another 24 h, the cells were incubated with 1 μg/ml Hoescht 33342 (Thermo) and 1:1000 CellMask Deep Red (Invitrogen) for 15 min at 37 °C. The cells were then fixed with 2.5% v/v glutaraldehyde and 2% v/v PFA in 0.1 M sodium cacodylate buffer, pH 7.4, at 37 °C for 30 min. The cells were imaged, in PBS, using a using a Zeiss LSM880 Airyscan microscope (Zeiss Airyscan detector, 63×/1.4 N oil immersion objectives), with care being taken to landmark the target cell on or near the Grid-50 pattern. The Airyscan images were deconvoluted in the Zeiss Zen Black software with the default Airyscan settings and then imported to ImageJ[65] where the images were saved as TIFF stacks. The TIFF stacks were denoised using the BM3D denoising algorithm in Python with a sigma psd of 0.01, and then upscaled by a factor of three using the bicubic algorithm and rotated and resliced, as needed, in ImageJ. Following imaging, the PBS was replaced with 2.5% v/v glutaraldehyde in 0.1 M sodium cacodylate buffer, pH 7.4, and the samples were stored at 4 °C until EM processing (approximately 4 days). For EM processing, the samples were first incubated in 2% w/v tannic acid (Electron Microscopy Sciences), 0.05 % w/v malachite green (Electron Microscopy Sciences), and 0.2% w/v ruthenium red (Electron Microscopy Sciences) in 0.1 M sodium cacodylate buffer, pH 7.4, for 1 h at RT. The cells were then washed three times with 0.1 M sodium cacodylate buffer, pH 7.4, and post-fixed with 1% w/v osmium tetroxide (Electron Microscopy Sciences) containing 0.2 % w/v ruthenium red in 0.1 M sodium cacodylate buffer, pH 7.4, for 1 h at RT. The cells were then washed three times with 0.1 M sodium cacodylate buffer, pH 7.4, and then three times with ddH$_2$O. The samples were then incubated in 0.5% w/v uranyl acetate (Electron Microscopy Sciences) overnight at 4 °C in the dark. The next morning, the samples were washed four times in ddH$_2$O and stepwise dehydrated in ethanol, followed by 100% acetonitrile. The samples were infiltrated with increasing mixtures of acetonitrile and Embed 812 resin (Electron Microscopy Sciences) until pure Embed resin was reached. The pure resin was exchanged three times and allowed to cure at 60 °C for four days. The samples were excised from the cover glass, trimmed to the target area, and mounted onto SEM stubs using colloidal silver (Electron Microscopy Sciences). A layer of 70 nm gold was added to the sample using a Leica EM Ace 200 Gold Sputterer prior to loading the sample into a Zeiss Crossbeam 550 operating under SmartSEM (Carl Zeiss Microscopy GmbH) and Atlas 3D (Fibics incorporated). Image voxel size was 7 nm in x/y and 10 nm in z. The resulting images were imported into Fiji v2.14.0 (ImageJ), where they were cropped and compiled into TIFF stacks. The stack alignment was refined using TrakEM2[66] and the images were denoised using a structure-preserving Gaussian denoising algorithm in Python[67]. Fluorescence and EM images were correlated using the Icy 2.4.3.0[68] plug-in ec-CLEMv2[69]. 3D modeling was performed using IMOD[70] and final image and movie compilations were created in Adobe Illustrator and Premiere Pro, respectively.

## Cell area measurements

Cells were seeded onto glass coverslips and transfected after 24 h, as described above, with a vector expressing GFP-only and the indicated myc-tagged RAB10 construct (if applicable). The next day, the cells were fixed with 2.5% v/v PFA (EM Sciences) in PBS for 10 min at 37 °C. Full fluorescent z-stacks were acquired using the Quorum spinning disk confocal microscope with a 40× NA 0.65 objective. Maximum intensity projections were created from the image stacks and analyzed using a custom script in Python, available at https://github.com/DrSydor/RAB10KO_CellArea (https://doi.org/10.5281/zenodo.10742961). Briefly, this script used a refined Cellpose[71,72] cytosol model to segment the cells and measured the areas of those cells in which the average

fluorescence intensity was above a pre-defined threshold. The results of this script were manually curated to remove any improperly segmented cells from the analysis.

## Atomic force microscopy

Henle 407 cells were plated at $1.0 \times 10^5$ cells per Willco dish (Warner Instruments) and transfected with peGFP-N1 plasmid. Twenty-four hours after transfection, all force-distance curves were collected using a Bioscope Resolve bioAFM system (Bruker) mounted on top of an Olympus IX-70 microscope base with Nanoscope software version 9.40. Regions of interest for the force spectroscopy experiments were identified optically using a 20× objective (UPlanFl, Olympus) and centered the AFM probe over individual cells. The force curves were acquired using pre-calibrated MLCT-D probes with a 45 µm beaded tip (Novascan Technologies). The tips were calibrated for deflection sensitivity on cell-free regions. The tip was positioned at the midpoint between the edge and the nucleus of the cells. To prevent cell damage, the tip was raised at least 30 µm from the cell surface before initiating force spectroscopy measurements. Force curves were measured with a ramp size of 3.0 µm and at a rate of 1 Hz, corresponding to a tip velocity of 6 µm/s. All analyses were performed on force curves collected with a deflection error trigger of 600 mV. All data collection took less than 1 h to ensure the health of the cells. Young's modulus values were determined by fitting the force curves using Indentation Analysis in Nano-Scope Analysis software version 1.90 with Contact Point Based fit and Hertzian (Spherical) model.

## Scanning electronic microscopy

Henle 407 cells were infected with WT *S*Tm and fixed with 2.5% $v/v$ glutaraldehyde in PBS, pH 7.4, 37 °C, for 2 h. Samples were dehydrated in an ethanol gradient (50%, 70%, and 90% $v/v$) for 15 min each, and then three exchanges with 100% ethanol for 15 min each. Dehydrated samples were dried in a Bal-Tec CPD030 critical point dryer (32 °C, 75 bar) and sputter coated with 10 nm gold in a Leica EM ACE200 high vacuum sputter coater. Imaging was done with a HITACHI FlexSEM 1000 II scanning electron microscope. The images were then imported into Fiji and the individual invasion ruffle (containing one or more bacteria) were manually drawn to create ROIs for area measurements. At least 15 invasion ruffles were counted in each condition and repeated for three independent experiments (at least 45 invasion ruffles in total for each condition).

## Statistics

Statistical analyses were conducted using GraphPad Prism v9.0. The mean +/− standard deviation (SD) is shown in the figures, and $P$ values were calculated using either an independent sample $t$ test, one-way ANOVA, two-way ANOVA, or two-tailed Mann–Whitney test. Tukey's HSD test was used as a post-hoc test for two-way ANOVA, as indicated in the figure legends. $P$ values were also included in corresponding graphs to denote the statistical significance.

## Reporting summary

Further information on research design is available in the Nature Portfolio Reporting Summary linked to this article.

## Data availability

The raw mass spectrometry data for the RAB10 BioID generated in this study has been deposited in the MassIVE public repository under accession # MSV000092081. Proximity interactions for FlagBirA*-RAB10 compared to 12× Flag-BirA* BioID alone runs performed under similar conditions (1% FDR) are presented in Supplementary Data 1. Source data are provided with this paper. Any additional information required to reanalyze the data reported in this paper is available from the lead contact upon request. Source data are provided with this paper.

## Code availability

Original Python code for the analysis of the cell areas is available at https://github.com/DrSydor/RAB10KO_CellArea (https://doi.org/10.5281/zenodo.10742961).

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

## Acknowledgements
J.H.B. holds the Pitblado Chair in Cell Biology. J.M. holds the GlaxoSmithKline Chair in Genetics and Genomics. A.M.M. holds a Canada Research Chair (Tier 1). Infrastructure for the Brumell Laboratory was provided by a John Evans Leadership Fund grant from the Canadian Foundation for Innovation and the Ontario Innovation Trust. This research was supported by operating grants from the Canadian Institutes of Health Research (FDN#154329 (J.H.B.), FDN#143202 (S.G.), MOP#142375 (J.M.), PJT#156093 (B.R.), FDN#408445 (A.M.M.)), the Natural Sciences and Engineering Research Council of Canada (C.M.Y.), the Leona M. and Harry B. Helmsley Charitable Trust (A.M.M.) and the National Institutes of Health Research (NIDDK RC2DK122532 (A.M.M.), RC2DK118640 (A.M.M.)). We thank Adam McCluggage for the generation of SopB mutant plasmids, Paul Paroutis for help with confocal microscopy, and Ali Darbandi for help with electron microscopy.

## Author contributions
Conceptualization: H.Z., K.C.B., J.M., A.M.S., J.H.B. Methodology: H.Z., A.M.S., K.C.B., A.A. Investigation: H.Z., A.M.S., K.C.B., A.A., B.-R.Y., E.C., E.M.N.L. Visualization: H.Z., A.M.S., K.C.B. Funding acquisition: J.H.B. Project administration: J.H.B., A.M.S. Supervision: J.H.B., A.M.M., B.R., C.M.Y., S.G. Writing—original draft: H.Z., J.M.J.T., J.H.B. Writing—review and editing: H.Z., J.M.J.T., A.M.S., K.C.B., A.A., A.M.M., B.R., C.M.Y., S.G., J.H.B.

## Competing interests
The authors declare no competing interests.
