## [Peer Review File · Nature Communications]

Salmonella exploits membrane reservoirs for invasion of host cellsREVIEWER COMMENTS

Reviewer #1 (Remarks to the Author):

Zhu et al, NCOMMS-23-42287-T

In this study Zhu et al investigated whether *Salmonella Typhimurium* (STm) can utilize membrane reservoirs for the generation of membrane ruffles in cultured epithelial cells. STm actively invade non-phagocytic cells using bacterial effector proteins delivered into the host cell via a Type III Secretion System (T3SS-1). This process has been extensively studied over more than 2 decades using cultured epithelial cells (e.g. HeLa and Henle 407). Ruffles can be very large, at least in vitro, but the source of membrane that feeds into the ruffles has not been identified. Here the authors address the role of intracellular Rab10-positive tubules in ruffle formation. The data suggest that these tubules are contiguous with the plasma membrane, appear to be important for maintaining the biophysical properties of the plasma membrane and are a source of membrane for STm-induced ruffles.

The conclusions are based primarily on quantification of Rab10+ tubules and invasion sites, from confocal images. However, I have concerns about the quality of the data. Specifically, it appears that many of the confocal images were analyzed without the use of Image analysis software and there is no explanation of how rab10+ tubules or invasion sites were defined. This is frankly unacceptable and surprising given that the authors used Image J to analyze ruffle size from SEM images. Please see the cited paper by Etoh & Fukuda, JCS, 2019 for quantitative analysis of tubules including colocalization of markers. Furthermore, it is impossible for the reader to estimate the variability within the system from the data shown, e.g. in Fig 1, 2 and 3. In comparison, the ruffle area quantification (Fig 4b) does at least show the variation (individual values are plotted rather than the mean of each experiment), although in this case the numbers of ruffles quantified are too low to provide a statistically robust analysis.

The results could certainly be of interest to the field but the authors should include in the discussion other relevant studies on the function of Rab10+ tubules, for example Kawai et al., 2021, *Front. Immunol.* 12:649600.

Other comments

- 1) The Introduction section (required for all Nature Communications articles) is absent.
- 2) Figures should be clearly labeled to clarify where time series of single cells are shown, especially in Fig 1. For example, "pre-infection" would be a more accurate and informative label than "uninfected". Also, where possible it would be helpful to have single color images in gray scale rather than green e.g. Fig 1b.
- 3) Error bars on graphs should show standard deviation (SD) rather than standard error of the mean (SEM), which gives no indication of the amount of variation in the system.
- 4) For the analysis of ruffles (Fig 4) the number of data points are surely too low for statistical analysis. For SEM "at least 5 ruffles of each condition were scored in each experiment" i.e. a total of 15 ruffles for WT. The fluorescence-based sampling was slightly better "at least 10 invasion ruffles of each condition were scored in each experiment" but it's hard to understand why such low numbers were assessed.

Reviewer #2 (Remarks to the Author):

Key results

The present manuscript by Zhu et al. reports on GFP-RAB10 localized to tubular structures that are disassembled by *Salmonella Typhimurium* infection. They demonstrate co-localization of different plasma membrane markers on these tubules and provide correlative fluorescence and FIB-SEM data to visualize the network of GFP-RAB10 tubules that reach the cell surface. A proximity-based screen (BioID) employing wt RAB10 and constitutively active and dominant negative RAB10

mutants revealed known interactors such as EHBP1 and MICAL-L1 as well as novel high-confidence interactors such as PACSIN2 and PACSIN3. They show localization of the Exocyst component EXOC2 and EXOC3 at STm invasion sites, which they next describe to originate from different cellular sites, i.e. EXOC2-RAB10 tubules and EXOC3-Cdc42-Golgi site. Next, by employing Δ SopB mutants that lack the binding capacity for CDC42, they provide data pointing to a relevance of the SopB-CDC42 axis for EXOC3 recruitment to invasion sites, but not EXOC2. Finally, the authors employ scanning EM to reveal a reduced ruffle area around STm invasion sites in RAB10, CDC42, EXOC2 and EXOC3 KOs.

Significance

The manuscript provides interesting new insights into the recruitment of cellular factors during Salmonella invasion. Technically elegant by employing FIB-SEM correlated to Airyscan fluorescent imaging, the authors convincingly show that over-expressed RAB10 localizes to tubules that are in continuum with the plasma membrane. The main novelty is assembled in Figure 3, which is the CDC42-dependent recruitment of EXOC3 to STm invasion sites, while EXOC2 is not recruited to these sites dependent on CDC42, and, by employing SopB mutants that were formally shown to be deficient for CDC42 binding, the authors show that EXOC3 is less efficiently recruited to these sites in this context. I mostly follow the line of arguments of localizations described (exceptions see points below). However, to justify publication of this manuscript, some critical experiments are missing in order to support the conclusions as drawn here, which I outline below.

Major points

1. In line 391, the authors state "Knockout efficiency was determined by western blot analysis." I am concerned about applying this technique without analyzing the gene. The CRISPR technique is available since many years and a proper genomic description of the genotype(s) is mandatory. Please provide genomic sequencing data for all knockouts, describe the type of genomic alteration (insertion, deletion, mutation) and show data for the number of alleles found. Otherwise, the description "knockout" is not valid. Please also provide Western Blots.
2. In line 140 ff, the authors conclude that RAB10 KO showing an increase in cell area are consistent with prior studies showing that PM invaginations serve as membrane reservoirs to control cell size. I cannot follow this argument. Firstly, the reference 7 does not show that membrane reservoirs are connected to control cell size. Please provide other references. Secondly, if a factor keeps lipid bilayers inside cells, such as put forward here for RAB10, deletion of this factor could also lead to fragmented membranes inside the cells or an increase of endosomes or other membrane structures. Please comment on this. Central to the conclusion of increased cell size is a reconstitution experiment: can the cell size be reverted by re-expression of RAB10? Are exocyst deficient cells also bigger?
3. Throughout the text, the use of overexpressed RAB10 (or other constructs) is not mentioned. Instead of "RAB10" it should read e.g. "GFP-RAB10" or "myc-PM-RAB10". It reads as if the function of endogenous RAB10 was addressed, which is not the case. It is well known that over-expression of constructs may lead to artefacts. So, the questions that I have are: does endogenous RAB10 also localize to tubules? Do RAB10 KO cells have these typical tubules? Otherwise, the wording that RAB10 is "actively" generating membrane reservoirs prior to infection as put forward in the title of the section in line 98 is an overstatement; it would rather be an over-expression artefact. Does the over-expression of RAB10 generate these tubules? If the latter is the case, this would imply a tubulation-inducing effect by over-expression of RAB10.
A subset of this point: In Figure 3b, myc-PM-RAB10 is used, while the authors refer to RAB10 in the text. Why do the authors use PM-RAB10 here and not unmodified RAB10? Please explain why the PM-version of RAB10 had to be used here or provide data on wt RAB10.
4. In Figure 3, the authors elaborate on EXOC2 and EXOC3. Firstly, synonyms of the Exocyst complex proteins (i.e. Sec5 and Sec6, respectively) are not introduced here. Secondly, the authors refer to "basal localization" (line 197) of EXOC2 and EXOC3. This is a wrong description of the data presented, since GFP-fusion proteins are over-expressed in Henle 407 cells. In that case, GFP-fusion proteins are constitutively expressed and most likely much more abundant as the endogenous protein. Please provide data for endogenous EXOC2 and EXOC3.
5. Related to Figure 3b, c: How does EXOC2 (and EXOC3) localize in uninfected cells, without overexpression of any RAB10?
6. Supplementary Figure S2a is too small and not readable, as well as most of the immunofluorescence images that are provided in post stamp size. Please provide readable Figures.

Minor points:

1. The first part of the manuscript presented here is a very close copy of a previously published paper by a subset of the same authors. In particular, the authors describe here: line 162 "Consistent with these findings, in WT Henle 407 cells we observed that EHBP1 and MICAL-L1 co-localize with RAB10 on membrane reservoirs ((Supplemental Fig. 2c)..." and in the Figure legend of Supplementary Figure 2: "c, Representative images of WT Henle 407 cells co-transfected with tdTomato-RAB10 (TdRAB10) and GFP-MICAL-L1 or GFP-EHBP1. White arrows indicate RAB10+ membrane reservoirs". In the paper by Boddy et al., NatCommun, 2021, <https://doi.org/10.1038/s41467-021-24983-z>, the authors have described the very same co-localization using the very same plasmids in the same cell line (Henle 407). In the Figure legend of this paper (Figure 4), it is described: "Representative images of Henle 407 live cells co-transfected with TD-RAB10 and GFP-MICAL-L1 or GFP-EHBP1. Arrowheads indicate colocalization on tubules." And the subsequent quantification of tubules in siRNA-treated cells is similarly already published in this paper. There is no reference to the own publication in this section. Instead, the authors reference to 2 papers describing the connection of RAB10 to EHBP1-EHD2 (ref19) and MICAL-L1 (ref20), respectively. This is misleading. Also, the localization and quantification of GFP-RAB10 positive invasion sites at 10 minutes post-invasion with STm and the SopB mutant (Figure 1c, d, e, f) is shown in the previous publication (Figure 2a, b Δ SopB). The reference to the own publication is missing in this paragraph. Please correct accordingly.
2. Localization of EXOC2 and EXOC3 to Salmonella invasion sites is not surprising, since the localization of the exocyst has been described more than 10 years ago. The paper describing exocyst localization (Nichols and Casanova, CurrBiol, 2010) has not been cited in the respective paragraph (line 208 ff). Please correct the paragraph and refer to published work.
3. One key type of analysis that is applied repeatedly throughout the manuscript is the quantification of co-localization with tubules and at invasion sites. However, how this quantification was done is neither described nor explained. Please provide exemplifying images on how this type of analysis was performed.
4. The second half of the manuscript is very difficult to read. Arguments and organization of figures is scattered; see e.g. the paragraph line 246 ff. It contains references to Figures 3k, S7a,b, 3h-j, S6, 3l,m, 3n, S8, S7c, d, while the section is just a quarter page. Please organize figures in a proper order. Moving supplementary data to the main may help.
5. The conclusion drawn in Supplementary Figure S5 on the relevance of SopE2 is at least over-interpreted and maybe wrong. The manuscript reads "Here, we observed that EXOC3 but not EXOC2 recruitment is inhibited during infection of Δ SopE2 mutant bacteria compared to WT STm (Supplemental Fig. 5)." However, Supplementary Figure S5 b,d shows indeed a statistical significant difference ($P < 0.05$), however, approx. 55% of EXOC2 and approx. 45% of EXOC3 are recruited to invasion sites. The statistical difference is likely to occur by different % localization of these proteins in the wt strain context: 65% of EXOC2 and 75% of EXOC3, respectively. In any case, to conclude an "inhibition" of recruitment from 30% reduction is wrong. Please re-word.
6. Along the same line of the point above, related to Supplementary Figure S5: the conclusion that "SopD, a SPI-1 T3SS effector with GAP activity towards RAB10, did not affect exocyst subunit recruitment (Supplemental Fig. 5)", while, in comparison, SopE2 would do, is over-interpreted. Both strains, Δ SopD and Δ SopE2, seem to behave very similar in terms of Exocyst complex localization. Even though statistical tests are provided, the differences between SopD and SopE2 are very mild. I cannot appreciate a difference in the images.
7. Figure 2c: What is the dashed box in the WGA PM dye supposed to show? There is no signal in the magenta channel.
8. Figure 2j, Cell area is labelled with μm^3 . I assume this labelling is wrong and should read μm^2 , otherwise the description "cell area" in the figure legend should read cell volume.
9. Supplementary Figure 2 Figures S2 e/f/g are placed in a wrong order: e/g/f, please correct
10. Wrong citations; in the context of Cdc42 interaction with SopB, Rab GTPases papers (Ref 39, 40) are cited. Please cite correctly, e.g. the structure of partial SopB with Cdc42 (Burkinshaw et al., JBC, 2012).
11. Line 266 ff: the relevance of CDC42 for ruffles has already been published and should be cited accordingly (Bosse et al., MBC, 2007).
12. References to STm mutant strains are missing in the material and methods section (lines 401, 406).

Reviewer #3 (Remarks to the Author):

The work Zhu and colleagues describes the novel phenomenon of plasma membrane (PM) invaginations that form a membrane reservoir in mammalian cells. In case of Salmonella invasion mediated by SPI-1 T3SS and its effector proteins, this membrane reservoir is rapidly recruited to sites of invasion to promote ruffle formation and bacterial uptake. Membrane reservoirs are associated with RAB10 and exocyst subunit EXOC2, and effector protein SopB mediates recruitment to sites of invasion. Independently SopB also mediates recruitment of EXOC3-positive membranes, enabling exocyst complex formation at sites of ruffle formation.

The study provides interesting new data, the experiments are in general well designed, and the presentation of the data is very good. Yet, I identified a number of points that should be addressed in order to improve the manuscript.

Specific comments:

1. The description of results and interpretation is rather brief, a proper discussion is missing. This probably results from prior submission to a journal with strict space limitation. This limitation is not given here, and a discussion should be included.
2. Prior work suggested infection-associated macropinosomes (IAMs) as reservoir for membrane material and contributing to increase of SCV surface. Is this a further mechanism that run in parallel to the PM tubules as membrane reservoir, or do the findings reported in this manuscript exclude a contribution of IAMs during Salmonella invasion?
3. Rab10-decorated tubules were observed in transfected cells expressing RAB10-GFP. This setting calls for a control to exclude that tubulation is a phenotype induced by RAB10 overexpression. Further experiments are required that demonstrate the existence of PM-derived tubules without RAB10 transfection. Perhaps the presence of CellMask as PM marker can be deployed, see experiment Fig. 2 f,g points. Is it possible to immunolabel endogenous RAB10 on CellMask-positive tubules?
4. The data shown in Fig. 4 address the effect of EXOC and RAB10 mutations on area and volume of membrane ruffles formed during Salmonella infection. Prior work indicated that membrane ruffles induced by Salmonella are highly dynamic and with morphology highly dependent on timepoint of analyses and number of Salmonella involved in the invasion event. The authors used for quantification ruffles at distinct timepoints after infection. With a low number of ruffles analysed in the study, this may not work. Either larger numbers of ruffles at single time points are analysed, or live-cell imaging is used for a smaller number of event to identify the maximal extension of ruffles and their quantification.
5. The model proposed in Fig. 4 f well summarized the reported findings. It also provokes new questions: If RAB10-associated PM invaginations are a general cellular membrane reservoir, they may also contribute to other process requiring massive membrane recruitment. I suggest to test internalization by other invasive pathogens for comparison an to test the generality of the observation. Obvious candidates are Shigella, Listeria, SPI-1 independent invasion of Salmonella mediated by Yersinia invasin, and perhaps Toxoplasma. Comparing some of these candidates could shed more light on the role of membrane reservoirs.
6. The work reports independent recruitment of EXOC2- and EXOC3-positive membranes to sites of invasion, following assembly of a functional exocyst complex. Is such individual recruitment of subunit also observed in other, non-pathogenic cellular process? What is the delivery route of the other subunits (only EXOC7 was mentioned)? To subcomplexes containing either EXOC2 or EXOC3, plus additional subunit pre-assemble, and than fuse to a functional exocyst complex?
7. That is the authors' explanation how SopB can act on two distinct compartments and direct to transport of EXOC2- as well as EXOC3-positive membranes to sites of invasion? Is this directed transport involving cytoskeletal filaments?

Reviewer #1 (Remarks to the Author):

In this study Zhu et al investigated whether Salmonella Typhimurium (STm) can utilize membrane reservoirs for the generation of membrane ruffles in cultured epithelial cells. STm actively invade non-phagocytic cells using bacterial effector proteins delivered into the host cell via a Type III Secretion System (T3SS-1). This process has been extensively studied over more than 2 decades using cultured epithelial cells (e.g. HeLa and Henle 407). Ruffles can be very large, at least in vitro, but the source of membrane that feeds into the ruffles has not been identified. Here the authors address the role of intracellular Rab10-positive tubules in ruffle formation. The data suggest that these tubules are contiguous with the plasma membrane, appear to be important for maintaining the biophysical properties of the plasma membrane and are a source of membrane for STm-induced ruffles.

The conclusions are based primarily on quantification of Rab10+ tubules and invasion sites, from confocal images. However, I have concerns about the quality of the data. Specifically, it appears that many of the confocal images were analyzed without the use of Image analysis software and there is no explanation of how rab10+ tubules or invasion sites were defined. This is frankly unacceptable and surprising given that the authors used Image J to analyze ruffle size from SEM images. Please see the cited paper by Etoh & Fukuda, JCS, 2019 for quantitative analysis of tubules including colocalization of markers. Furthermore, it is impossible for the reader to estimate the variability within the system from the data shown, e.g. in Fig 1, 2 and 3. In comparison, the ruffle area quantification (Fig 4b) does at least show the variation (individual values are plotted rather than the mean of each experiment), although in this case the numbers of ruffles quantified are too low to provide a statistically robust analysis.

Response: We are sincerely grateful for the thoughtful suggestions and guidance provided by the reviewer. In response to their valuable feedback, we have incorporated comprehensive details on the methods employed for imaging analysis and data presentation in the 'Materials and Methods' section of the revised manuscript.

Specifically, we manually identified and measured the number of cells containing at least one RAB10⁺ tubule (or one CellMask⁺ tubule >10 μm in length if specified, as in Fig. 3c,d in the revised manuscript), aligning with the method outlined by Etoh & Fukuda, 2019 (PMID: 30700496). An enrichment of the indicated protein's signal at the STm invasion site, relative to the signal in the cytosol, was considered a positive recruitment, in accordance with Boddy et al., 2021 (PMID: 34349110). Colocalizations were determined using Mander's Correlation Coefficient (MCC) values (which we believe better represents the colocalizations on certain compartments than PCC in this context) calculated on Regions of Interest (ROI) through Coloc 2 in Fiji (ImageJ), following the methodology outlined by Etoh & Fukuda, 2019 (PMID: 30700496). To further support the colocalizations, line plot profiles were obtained using the plot profile function in Fiji, as described in Etoh & Fukuda, 2019 (PMID: 30700496). Other necessary details regarding image analysis can be found in the 'Materials and Methods' section.

To Address the question regarding ruffle area measurement in SEM images, as suggested by the reviewer and recognizing the importance of sample size in statistical analyses, we have increased the sample size of SEM-based 2D measurement of ruffle area from 15 to a total of 45 ruffles for each condition (Fig. 7a,b) in the revised manuscript, aiming to enhance the robustness of our data. We also acknowledge the limitations of 2D measurements manually using Fiji (ImageJ) which only partially reflects the difference in the ruffling abilities of different cell lines. To address this, we have incorporated 3D volume measurements at multiple time points with fluorescence labeling to more comprehensively assess ruffling abilities (Fig. 7c,d in the revised manuscript).

For clarity in data presentation, in Fig. 1, 2, and 3 in the original manuscript, each dot in the graph represents the data (not mean value) from each biological replicate. These dots usually represent manually counted cells with membrane reservoirs out of all cells, or invasion sites with the indicated protein recruitment out of all invasion sites. The number is presented as a percentage (not mean value) for each biological replicate, and the variation among biological replicates (3 to 6 replicates) is represented with error bars. For experiments providing absolute values, For example, Fig. 3e,g & Fig. 7b in the revised manuscript, individual values and error bars are plotted to depict the variation.

The results could certainly be of interest to the field but the authors should include in the discussion other relevant studies on the function of Rab10+ tubules, for example Kawai et al., 2021, Front. Immunol. 12:649600.

Response: We appreciate the reviewer's observation regarding the multifunctionality of RAB10 in various intracellular compartments, including RAB10⁺ tubular structures. In response to this insightful consideration, we have expanded our discussions on this topic in the 'Discussion' section of the revised manuscript. We believe that these additional insights contribute to a more comprehensive understanding of RAB10's diverse roles within the cell.

Other comments

1) The Introduction section (required for all Nature Communications articles) is absent.

Response:

We express our gratitude to the reviewer for bringing this to our attention. In response, we have integrated the 'Introduction' section into the revised manuscript. Additionally, we have updated other formatting elements in alignment with the guidelines provided by *Nature Communications*.

2) Figures should be clearly labeled to clarify where time series of single cells are shown, especially in Fig 1. For example, “pre-infection” would be a more accurate and informative label than “uninfected”. Also, where possible it would be helpful to have single color images in gray scale rather than green e.g. Fig 1b.

Response: We thank the reviewer for the suggestion regarding labeling. In response, we have made revisions to the labeling in the time series images of Figure 1, specifically changing 'Uninfected' to 'Pre-infection.' Furthermore, in Figure 1b, we have adjusted the color of STm from blue to red to enhance its visibility. These modifications aim to improve clarity and better address the concerns raised.

3) *Error bars on graphs should show standard deviation (SD) rather than standard error of the mean (SEM), which gives no indication of the amount of variation in the system.*

Response: In response to the reviewer's suggestion, we have implemented changes in all graphs, replacing standard error of the mean (SEM) with standard deviation (SD) for the error bars. Additionally, we have now included P values directly on the graphs to present the statistical information in a more straightforward manner.

4) *For the analysis of ruffles (Fig 4) the number of data points are surely too low for statistical analysis. For SEM "at least 5 ruffles of each condition were scored in each experiment" i.e. a total of 15 ruffles for WT. The fluorescence-based sampling was slightly better "at least 10 invasion ruffles of each condition were scored in each experiment" but it's hard to understand why such low numbers were assessed.*

Response: We thank the reviewer for the suggestion. As addressed in the response to Comment #1, we have expanded the sample sizes as recommended.

Reviewer #2 (Remarks to the Author):

Key results

The present manuscript by Zhu et al. reports on GFP-RAB10 localized to tubular structures that are disassembled by Salmonella Typhimurium infection. They demonstrate co-localization of different plasma membrane markers on these tubules and provide correlative fluorescence and FIB-SEM data to visualize the network of GFP-RAB10 tubules that reach the cell surface. A proximity-based screen (BioID) employing wt RAB10 and constitutively active and dominant negative RAB10 mutants revealed known interactors such as EHBP1 and MICAL-L1 as well as novel high-confidence interactors such as PACSIN2 and PACSIN3. They show localization of the Exocyst component EXOC2 and EXOC3 at STm invasion sites, which they next describe to originate from different cellular sites, i.e. EXOC2-RAB10 tubules and EXOC3-Cdc42-Golgi site. Next, by employing Δ SopB mutants that lack the binding capacity for CDC42, they provide data pointing to a relevance of the SopB-CDC42 axis for EXOC3 recruitment to invasion sites, but not EXOC2. Finally, the authors employ scanning EM to reveal a reduced ruffle area around STm invasion sites in RAB10, CDC42, EXOC2 and EXOC3 KOs.

Significance

The manuscript provides interesting new insights into the recruitment of cellular factors during

Salmonella invasion. Technically elegant by employing FIB-SEM correlated to Airyscan fluorescent imaging, the authors convincingly show that over-expressed RAB10 localizes to tubules that are in continuum with the plasma membrane. The main novelty is assembled in Figure 3, which is the CDC42-dependent recruitment of EXOC3 to STm invasion sites, while EXOC2 is not recruited to these sites dependent on CDC42, and, by employing SopB mutants that were formally shown to be deficient for CDC42 binding, the authors show that EXOC3 is less efficiently recruited to these sites in this context. I mostly follow the line of arguments of localizations described (exceptions see points below). However, to justify publication of this manuscript, some critical experiments are missing in order to support the conclusions as drawn here, which I outline below.

Major points

1. In line 391, the authors state “Knockout efficiency was determined by western blot analysis.” I am concerned about applying this technique without analyzing the gene. The CRISPR technique is available since many years and a proper genomic description of the genotype(s) is mandatory. Please provide genomic sequencing data for all knockouts, describe the type of genomic alteration (insertion, deletion, mutation) and show data for the number of alleles found. Otherwise, the description “knockout” is not valid. Please also provide Western Blots.

Response: Apologies for the oversight in the initial submission; unfortunately, the western blots were unintentionally omitted. In this revised version, we have included the western blot data for all four knockout cell lines used in this study (Supplemental Fig. 2). It's worth noting that two of these cell lines (*RAB10* KO cells in Boddy et al., 2021, PMID: 34349110; *CDC42* KO cells in Walpole et al., 2022 PMID 35484249) have been previously described and verified in other studies, and the same cells were employed in this study. Additionally, the *EXOC2* and *EXOC3* KO cell lines were specifically generated for this study. We conducted sequencing for all four cell lines, comparing them with wild-type sequences to identify genomic alterations and allele information (Supplemental Fig. 2).

Regarding the *CDC42* KO cells, we observed that it is not a monoallelic mutation from the sequencing data. However, we have verified the depletion of protein expression by Western Blot both in our project and in a prior study (Walpole et al., 2022 PMID 35484249). Consequently, we believe that, within the scope of this study, the *CDC42* KO cell line still serves as a suitable platform for exploring *CDC42* functions.

2. In line 140 ff, the authors conclude that *RAB10* KO showing an increase in cell area are consistent with prior studies showing that PM invaginations serve as membrane reservoirs to control cell size. I cannot follow this argument. Firstly, the reference 7 does not show that membrane reservoirs are connected to control cell size. Please provide other references. Secondly, if a factor keeps lipid bilayers inside cells, such as put forward here for *RAB10*, deletion of this factor could also lead to fragmented membranes inside the cells or an increase of endosomes or other membrane structures. Please comment on this. Central to the conclusion of increased cell size is a reconstitution experiment: can the cell size be reverted by re-expression of *RAB10*? Are exocyst deficient cells also bigger?

Response: We appreciate the reviewer for bringing up this point. Indeed, we acknowledge that the statement regarding "controlling cell size" may not be precisely accurate. Instead, as demonstrated in reference #7, the membrane reservoirs have been shown to regulate the cell's PM surface area. Therefore, we have modified the text to be more accurate, see Line 150 in the revised manuscript (This and the following references to line numbers pertain to the revised version of the manuscript with 'Track Changes' turned off, as the line numbering is discontinuous in the other revised manuscript file with tracked changes over the original one). In our study, we quantified the maximum peripheral cell area as an indicator of the PM area (refer to the "Materials and Methods" section for details).

We have now incorporated additional data of the reconstitution experiment, revealing that the knockout of *RAB10* led to an increase in cell PM area. Notably, the expression of either WT *RAB10* or PM-targeted *RAB10* successfully rescued this phenotype. These findings strongly suggest that *RAB10* actively regulates the PM area.

In response to the reviewer's comment on the potential membrane fragmentation due to *RAB10* depletion, our current data indicates that *RAB10* plays a crucial role in the formation of deep/long PM⁺ reservoirs. The deletion of *RAB10* hinders the initial development of these membrane reservoirs (Fig. 3c,d in the revised manuscript). Therefore, we anticipate that in cells depleted of *RAB10*, there would be no additional generation of fragmented membrane structures within the cell or an increase in endosomal or other membrane structures. However, it is important to note that further investigations are necessary to validate this hypothesis. We appreciate this insightful comment from the reviewer, as they provide valuable considerations for future directions in our research. Exploring the potential impact of *RAB10* depletion on membrane dynamics is an intriguing avenue that we look forward to exploring in subsequent studies.

Regarding cells with lacking expression of specific exocyst subunits (*EXOC2* or *EXOC3*), our preliminary observations indicate that these cells displayed similar sizes and cell PM areas as WT cells. This aligns with the data suggesting that *EXOC2* KO cells still maintain *RAB10*⁺ membrane reservoirs (Supplemental Fig. 5e,f in the revised manuscript). Consequently, we think that neither *EXOC2* nor *EXOC3* (which is not localized to *RAB10*⁺ tubules at all) significantly contribute to reservoir formation, and do not play a substantial role in controlling PM area. However, we acknowledge that the potential role of the exocyst complex in regulating cell area remains to be determined in future studies.

3. Throughout the text, the use of overexpressed RAB10 (or other constructs) is not mentioned. Instead of "RAB10" it should read e.g. "GFP-RAB10" or "myc-PM-RAB10". It reads as if the function of endogenous RAB10 was addressed, which is not the case. It is well known that over-expression of constructs may lead to artefacts. So, the questions that I have are: does endogenous RAB10 also localize to tubules? Do RAB10 KO cells have these typical tubules? Otherwise, the wording that RAB10 is "actively" generating membrane reservoirs prior to

infection as put forward in the title of the section in line 98 is an overstatement; it would rather be an over-expression artefact. Does the over-expression of RAB10 generate these tubules? If the latter is the case, this would imply a tubulation-inducing effect by over-expression of RAB10. A subset of this point: In Figure 3b, myc-PM-RAB10 is used, while the authors refer to RAB10 in the text. Why do the authors use PM-RAB10 here and not unmodified RAB10? Please explain why the PM-version of RAB10 had to be used here or provide data on wt RAB10.

Response: We thank the reviewer pointing this out. We fully concur with the reviewer's consideration that overexpression may introduce artifacts.

In response to the reviewer's query regarding the localization of endogenous RAB10 on tubules, we have now incorporated immunofluorescence staining images of endogenous RAB10 with the PM marker (PM-mCherry) in Figure 2f,g (shown below). Following quantification, approximately 40% of untransfected WT Henle cells exhibit these endogenous RAB10⁺ (PM⁺)tubules.

It is worth noting that our preference for overexpressing RAB10 constructs (GFP-RAB10 & myc-PM-RAB10) instead of relying solely on immunofluorescence staining of endogenous RAB10 is attributed to several reasons: (1) The PM dye, CellMask, widely employed in our study is not compatible with immunofluorescence staining after cell membrane permeabilization; (2) Our specific interest in RAB10 on the PM led us to use myc-PM-RAB10 to exclude RAB10 from other cellular compartments; (3) The use of fluorescence-tagged RAB10 constructs facilitates live-cell imaging or imaging with CellMask directly after fixation. Importantly, we have confirmed the functionality of the RAB10 constructs utilized in our study (Fig. 3d,g in the revised manuscript).

Regarding the reviewer's question about the presence of typical tubules in *RAB10* KO cells, we have included data demonstrating that the knockout of *RAB10* eliminates the PM tubules marked by CellMask (Fig. 3c,d in the revised manuscript).

Therefore, these data above support the conclusion that endogenous RAB10 is capable of generating these PM-associated tubular reservoirs.

In response to the reviewer's query regarding Figure 3b (Figure 5b in the revised manuscript), we have incorporated additional data for PM-targeted RAB10. This inclusion is intended to provide further support for the proposition that EXOC2 predominantly localizes to RAB10⁺ PM-associated membrane reservoirs rather than other intracellular compartments that contain RAB10. Recognizing the merit of the reviewer's suggestion to enhance the robustness of our conclusions, we have now incorporated data for WT RAB10 in Supplemental Fig. 5d,g.

4. In Figure 3, the authors elaborate on EXOC2 and EXOC3. Firstly, synonyms of the Exocyst complex proteins (i.e. Sec5 and Sec6, respectively) are not introduced here. Secondly, the authors refer to "basal localization" (line 197) of EXOC2 and EXOC3. This is a wrong description of the data presented, since GFP-fusion proteins are over-expressed in Henle 407 cells. In that case, GFP-fusion proteins are constitutively expressed and most likely much more abundant as the endogenous protein. Please provide data for endogenous EXOC2 and EXOC3.

Response: We thank the reviewer for pointing this out. We now incorporate the synonyms in the revised manuscript. Additionally, we have replaced the term "basal localization" with "localization prior to STm infection" to more accurately convey the specific localization context of EXOC2 and EXOC3.

Regarding the reviewer's inquiry about the endogenous localization of EXOC2 and EXOC3 in Henle 407 cells, our initial plan involved using immunofluorescence staining with antibodies. Unfortunately, we encountered challenges in obtaining reliable signals with these antibodies when comparing WT cells with *EXOC2* KO or *EXOC3* KO cells, as illustrated in the figure below. As an alternative, we opted for fluorescence-tagged constructs of EXOC2 and EXOC3, a method commonly employed in prior studies and validated as functional in various contexts, including studies of STm infection (see, for example, PMID: 20579884, 33599020, 35151976, etc.). This approach was chosen to ensure the robustness and reliability of our experimental observations.

Primary antibody:

& : rabbit polyclonal anti-EXOC3 (Proteintech #14703-1-AP)
 # : rabbit monoclonal anti-EXOC2 (Abcam # ab140620)

5. Related to Figure 3b, c: How does EXOC2 (and EXOC3) localize in uninfected cells, without overexpression of any RAB10?

Response: We appreciate the reviewer's understanding, and we apologize for not including EXOC2's and EXOC3's localizations in uninfected cells without RAB10 overexpression in the original manuscript. As depicted in the figure below, we have now included this data in Supplemental Fig. 5c. Specifically, EXOC2 is observed to localize to tubular structures, while EXOC3 exhibits localization to peri-nuclear regions. These findings align with the rationale behind subsequently testing their co-localizations with RAB10⁺ tubules and Golgi marker GM130.

6. Supplementary Figure S2a is too small and not readable, as well as most of the

immunofluorescence images that are provided in post stamp size. Please provide readable Figures.

Response: We thank the reviewer for the suggestion. In response, we have increased the sizes of relevant images to enhance the overall reading experience.

Minor points:

1. The first part of the manuscript presented here is a very close copy of a previously published paper by a subset of the same authors. In particular, the authors describe here: line 162 “Consistent with these findings, in WT Henle 407 cells we observed that EHBP1 and MICAL-L1 co-localize with RAB10 on membrane reservoirs ((Supplemental Fig. 2c)...” and in the Figure legend of Supplementary Figure 2: “c, Representative images of WT Henle 407 cells co-transfected with tdTomato-RAB10 (TdRAB10) and GFP-MICAL-L1 or GFP-EHBP1. White arrows indicate RAB10+ membrane reservoirs”. In the paper by Boddy et al., NatCommun, 2021, <https://doi.org/10.1038/s41467-021-24983-z>, the authors have described the very same co-localization using the very same plasmids in the same cell line (Henle 407). In the Figure legend of this paper (Figure 4), it is described: “Representative images of Henle 407 live cells co-transfected with TD-RAB10 and GFP-MICAL-L1 or GFP-EHBP1. Arrowheads indicate colocalization on tubules.” And the subsequent quantification of tubules in siRNA-treated cells is similarly already published in this paper. There is no reference to the own publication in this section. Instead, the authors reference to 2 papers describing the connection of RAB10 to EHBP1-EHD2 (ref19) and MICAL-L1 (ref20), respectively. This is misleading. Also, the localization and quantification of GFP-RAB10 positive invasion sites at 10 minutes post-invasion with STm and the SopB mutant (Figure 1c, d, e, f) is shown in the previous publication (Figure 2a, b Δ SopB). The reference to the own publication is missing in this paragraph. Please correct accordingly.

Response: We appreciate the reviewer's understanding and acknowledgment of the clarification provided. We apologize for any confusion caused. In the study by Boddy et al., 2021 (PMID: 34349110), the co-localization of EHBP1 and MICAL-L1 with RAB10+ tubules was indeed demonstrated (Fig. 4a in Boddy et al., 2021, PMID: 34349110). In our original manuscript, Supplemental Fig. 2c served as supporting data for quantifications, but we recognize that it may have appeared repetitive. As per the reviewer's suggestion, we have cited the relevant data in Boddy et al., 2021 and removed the redundant images.

Regarding the reference to SopB-mediated RAB10 recruitment, we have described and cited the findings from the prior study (Boddy et al., 2021, PMID: 34349110) in the paragraph earlier (Line 80-81 in the revised manuscript, Line 82-83 in the original manuscript). The focus of Fig. 1c-f in this current manuscript is primarily on the phenotypes associated with SopB-driven concomitant disassembly of RAB10+ membrane reservoirs and RAB10 delivery to STm invasion sites. Besides, these observations are supported by data obtained at multiple time points. We hope this clarification addresses the concerns raised by the reviewer.

2. Localization of EXOC2 and EXOC3 to Salmonella invasion sites is not surprising, since the localization of the exocyst has been described more than 10 years ago. The paper describing exocyst localization (Nichols and Casanova, CurrBiol, 2010) has not been cited in the respective paragraph (line 208 ff). Please correct the paragraph and refer to published work.

Response: We sincerely appreciate the reviewer's acknowledgment of the seminal work by Nichols and Casanova, which has inspired our research here. We totally agree with the importance of their pioneering discovery, as highlighted in their publication in Current Biology, 2010 (PMID: 20579884). In our manuscript, we have cited this work in an earlier paragraph (Line 198-199 in the revised manuscript, Line 186-187 in the original manuscript) to underscore their contributions.

It is worth noting that Nichols and Casanova primarily investigated the recruitment of a few exocyst subunits to Salmonella Typhimurium (STm) invasion sites, with a focus on subunits other than EXOC3 (Sec6). In our effort, we included data on EXOC3 and two additional exocyst subunits' recruitment (Fig. 5a & Supplemental Fig. 5a in the revised manuscript, Fig. 3a & Supplemental Fig. 3a in the original manuscript).

Regarding the reviewer's comment on the reference at Line 208 (in the original manuscript), we appreciate the guidance provided. In response, we have added a sentence of "Previous study² and our data above identified exocyst subunit recruitments to STm invasion sites,..."(Line 224-225 in the revised manuscript) to explicitly recognize Nichols and Casanova's work and emphasize the identification of exocyst subunits' recruitments to STm invasion sites in both previous studies and our current findings.

3. One key type of analysis that is applied repeatedly throughout the manuscript is the quantification of co-localization with tubules and at invasion sites. However, how this quantification was done is neither described nor explained. Please provide exemplifying images on how this type of analysis was performed.

Response: As addressed in the response to Comment #1 of Reviewer 1 above, we have taken the feedback into careful consideration and have now provided comprehensive details on image analysis in the "Materials and Methods" section. This includes a thorough explanation of the methodologies employed for colocalization, such as MCC values and Line plot profiles, as well as the identification and quantification of recruitment at invasion sites. Additionally, we have outlined all other measurements utilized in this study.

4. The second half of the manuscript is very difficult to read. Arguments and organization of figures is scattered; see e.g. the paragraph line 246 ff. It contains references to Figures 3k, S7a,b, 3h-j, S6, 3l,m, 3n, S8, S7c, d, while the section is just a quarter page. Please organize figures in a proper order. Moving supplementary data to the main may help.

Response: We thank the reviewer for the suggestion. In response, we have reorganized the figures by expanding the major display to include 7 figures and splitting the supplementary figures accordingly. Our aim is to enhance the overall reading experience and make the presentation more accessible.

5. *The conclusion drawn in Supplementary Figure S5 on the relevance of SopE2 is at least over-interpreted and maybe wrong. The manuscript reads “Here, we observed that EXOC3 but not EXOC2 recruitment is inhibited during infection of Δ sopE2 mutant bacteria compared to WT STm (Supplemental Fig. 5).” However, Supplementary Figure S5 b,d shows indeed a statistical significant difference ($P < 0.05$), however, approx. 55% of EXOC2 and approx. 45% of EXOC3 are recruited to invasion sites. The statistical difference is likely to occur by different % localization of these proteins in the wt strain context: 65% of EXOC2 and 75% of EXOC3, respectively. In any case, to conclude an “inhibition” of recruitment from 30% reduction is wrong. Please re-word.*

Response: To address this concern, we have revised the wording to more accurately describe the SopE2 phenotype. Specifically, we have replaced "inhibited" with "diminished" (Line 242 in the revised manuscript).

6. *Along the same line of the point above, related to Supplemental Figure S5: the conclusion that “SopD, a SPI-1 T3SS effector with GAP activity towards RAB10, did not affect exocyst subunit recruitment (Supplemental Fig. 5)”, while, in comparison, SopE2 would do, is over-interpreted. Both strains, Δ SopD and Δ SopE2, seem to behave very similar in terms of Exocyst complex localization. Even though statistical tests are provided, the differences between SopD and SopE2 are very mild. I cannot appreciate a difference in the images.*

Response: We appreciate the reviewer's insightful suggestion, and in response, we have incorporated a more detailed description of how the recruitment at invasion sites was identified and counted. While we did observe a decrease in EXOC3 recruitment in cells infected with Δ sopD STm, the statistical analysis indicated that this decrease is not significant. To provide a more precise description of the phenotype, we have updated the text to state, "*SopD, a SPI-1 T3SS effector with GAP activity towards RAB10⁴, did not significantly affect exocyst subunit recruitment*" (see Line 247-249 of the revised manuscript). Furthermore, we have replaced the representative images of Δ sopD STm infection in Supplemental Fig. 5c in the original manuscript (now Supplemental Fig. 7a-d in the revised manuscript).

7. *Figure 2c: What is the dashed box in the WGA PM dye supposed to show? There is no signal in the magenta channel.*

Response: We have now made sure that the signal in magenta channel can be clearly identified. Additionally, we have included an inset of the region of interest (ROI) within the dashed box, accompanied by its corresponding line plot profile.

8. *Figure 2j, Cell area is labelled with μm^3 . I assume this labelling is wrong and should read μm^2 , otherwise the description “cell area” in the figure legend should read cell volume.*

Response: We thank the reviewer for pointing this out. We have corrected the labelling of cell area to μm^2 .

9. *Supplementary Figure 2 Figures S2 e/f/g are placed in a wrong order: e/g/f, please correct*

Response: Supplemental Fig. 2 e-g have been moved to display figures and the order corrected.

10. *Wrong citations; in the context of Cdc42 interaction with SopB, Rab GTPases papers (Ref 39, 40) are cited. Please cite correctly, e.g. the structure of partial SopB with Cdc42 (Burkinshaw et al., JBC, 2012).*

Response: We thank the reviewer for pointing this out. We have rectified the issue by citing the correct reference.

11. *Line 266 ff: the relevance of CDC42 for ruffles has already been published and should be cited accordingly (Bosse et al., MBC, 2007).*

Response: We appreciate the suggestion from the reviewer, and in response, we have now cited and provided a description of the work regarding CDC42's established role in *Listeria* ruffle formation (Line 284-285 in the revised manuscript).

12. *References to STm mutant strains are missing in the material and methods section (lines 401, 406).*

Response: We appreciate the reviewer's diligence, and in response to the feedback, we have updated the "Material and Methods" section to include appropriate strain information.

Reviewer #3 (Remarks to the Author):

The work Zhu and colleagues describes the novel phenomenon of plasma membrane (PM) invaginations that form a membrane reservoir in mammalian cells. In case of Salmonella invasion mediated by SPI-1 T3SS and its effector proteins, this membrane reservoir is rapidly recruited to sites of invasion to promote ruffle formation and bacterial uptake. Membrane reservoirs are associated with RAB10 and exocyst subunit EXOC2, and effector protein SopB mediates recruitment to sites of invasion. Independently SopB also mediates recruitment of EXOC3-positive membranes, enabling exocyst complex formation at sites of ruffle formation.

The study provides interesting new data, the experiments are in general well designed, and the presentation of the data is very good. Yet, I identified a number of points that should be addressed in order to improve the manuscript.

Specific comments:

1. *The description of results and interpretation is rather brief, a proper discussion is missing. This probably results from prior submission to a journal with strict space limitation. This limitation is not given here, and a discussion should be included.*

Response: We are grateful for the reviewer's suggestion. To address it, we have incorporated "Introduction" and "Discussion" sections following the formatting guidelines of *Nature Communications*. Additionally, we have taken steps to enhance the clarity of our results by reorganizing and adding pertinent text.

2. *Prior work suggested infection-associated macropinosomes (IAMs) as reservoir for membrane material and contributing to increase of SCV surface. Is this a further mechanism that run in parallel to the PM tubules as membrane reservoir, or do the findings reported in this manuscript exclude a contribution of IAMs during Salmonella invasion?*

Response: We appreciate the significance of the findings regarding the role of IAMs in STm intracellular growth by providing membranes for SCVs. We agree that the cooperative action of IAMs formed upon STm infection, along with RAB10⁺ membrane reservoirs and exocyst delivery, could cooperatively contribute to early invasion events such as ruffling and PM scission. In response to this insightful comment, we have now cited and discussed related work in the "Discussion" section of the revised manuscript.

3. *Rab10-decorated tubules were observed in transfected cells expressing RAB10-GFP. This setting calls for a control to exclude that tubulation is a phenotype induced by RAB10 overexpression. Further experiments are required that demonstrate the existence of PM-derived tubules without RAB10 transfection. Perhaps the presence of CellMask as PM marker can be deployed, see experiment Fig. 2 f,g points. Is it possible to immunolabel endogenous RAB10 on CellMask-positive tubules?*

Response: We thank the reviewer for pointing this out. As mentioned in our earlier responses to Comment #3 of Reviewer 2, we have now included immunolabeling images of endogenous RAB10 with the PM marker (PM-mCherry). It's important to note that CellMask staining does not work after permeabilization, which is a necessary step for the immunolabeling of endogenous RAB10. As an alternative, we utilized PM-mCherry as the PM marker in this context.

4. *The data shown in Fig. 4 address the effect of EXOC and RAB10 mutations on area and volume of membrane ruffles formed during Salmonella infection. Prior work indicated that membrane ruffles induced by Salmonella are highly dynamic and with morphology highly dependent on timepoint of analyses and number of Salmonella involved in the invasion event. The authors used for quantification ruffles at distinct timepoints after infection. With a low number of ruffles analysed in the study, this may not work. Either larger numbers of ruffles at single time points are analysed, or live-cell imaging is used for a smaller number of event to identify the maximal extension of ruffles and their quantification.*

Response: In response to the reviewer's inquiry regarding the measurement of ruffle area in SEM images, we fully acknowledge the limitations of measurement of invasion ruffle at certain timepoints (10 min p.i. was chosen in this study), as it may only provide a partial representation of variations in the ruffling capabilities of distinct cells. Consequently, we have also incorporated 3D volume measurements at multiple time points using fluorescence labeling (Fig. 7c,d in the revised manuscript).

As the reviewer suggested, we expanded the sample size enhances the statistical robustness of our analysis. For SEM-based 2D ruffle area measurement, we have now augmented the sample size to a total of 45 ruffles for each condition. For live-cell 3D ruffle volume measurement, we included a total of 45 ruffles for each condition.

5. The model proposed in Fig. 4 f well summarized the reported findings. It also provokes new questions: If RAB10-associated PM invaginations are a general cellular membrane reservoir, they may also contribute to other process requiring massive membrane recruitment. I suggest to test internalization by other invasive pathogens for comparison an to test the generality of the observation. Obvious candidates are Shigella, Listeria, SPI-1 independent invasion of Salmonella mediated by Yersinia invasin, and perhaps Toxoplasma. Comparing some of these candidates could shed more light on the role of membrane reservoirs.

Response: We recognize the importance of extending our investigation into STm to encompass other biological contexts that involve substantial membrane recruitment, such as the infection caused by different other bacterial pathogens. This indeed represents a significant avenue for future research. However, as exemplified in our current study, such future studies will require comprehensive efforts to characterize the dynamics of membrane reservoirs during infection and the mechanisms by which microbial virulence factors can modulate them. Thus, we believe our manuscript will provide an important 'roadmap' for future studies of other pathogens by members of the microbial pathogenesis field.

6. The work reports independent recruitment of EXOC2- and EXOC3-positive membranes to sites of invasion, following assembly of a functional exocyst complex. Is such individual recruitment of subunit also observed in other, non-pathogenic cellular process? What is the delivery route of the other subunits (only EXOC7 was mentioned)? To subcomplexes containing either EXOC2 or EXOC3, plus additional subunit pre-assemble, and than fuse to a functional exocyst complex?

Response: We appreciate the reviewer's insight regarding the significance of exploring the delivery route and assembly pattern of exocyst subunits. This is indeed an important and intriguing topic for future research, both within the context of bacterial infection and in non-pathogenic cellular processes.

As outlined in the response to comment #5 above, we are committed to investigate the role of the exocyst complex (individual subunits and the holo-complex) in various processes crucial for maintaining plasma membrane homeostasis. In line with the

reviewer's suggestion, we have included mention of some of the future directions in the "Discussion" section (Line 334-348 in the revised manuscript), emphasizing the need to further investigate these aspects.

7. That is the authors' explanation how SopB can act on two distinct compartments and direct to transport of EXOC2- as well as EXOC3-positive membranes to sites of invasion? Is this directed transport involving cytoskeletal filaments?

Response: We sincerely appreciate the reviewer's insightful suggestion. SopB is a versatile bacterial effector that keeps fascinating us: For example, SopB facilitates actin remodeling at the invasion site by promoting Annexin 2 (AnxA2) recruitment, serving as a platform for actin rearrangements (Jolly et al., 2014, PMID: 23931152). Additionally, SopB aids in RhoJ recruitment to the invasion site, contributing to bacterial internalization through an undefined mechanism (Truong et al., 2018, PMID: 36717589). SopB promotes cytosolic SNX18 recruitment to the invasion site to facilitate SCV formation and scission, via its lipid phosphatase activity (Liebl et al., 2017, PMID: 28664153). SopB is also found to direct R-Ras1, RhoB, and RhoH to the invasion site, contributing to SopB-mediated Akt activation (Truong et al., 2018, PMID: 36717589). These diverse signalling events were found mostly initiated by SopB's lipid phosphatase activity, or by mechanisms undefined yet.

In this current study, we have dissected the mechanisms of SPI-1 T3SS effectors' role (especially SopB) in regulating the membrane reservoir structures, host GTPases and exocyst subunits. The interplay between the pathogen and the host is intricate, and it is likely that additional mechanisms govern SopB-mediated delivery of membrane and exocyst subunits from distinct cellular compartments. Our characterization in this study has focused on SopB's lipid phosphatase activity and CDC42 binding, enabling the recruitment of RAB10/EXOC2 and CDC42/EXOC3 independently. However, we acknowledge that further complexities and mechanisms may contribute to this intricate interplay.

We acknowledge the importance highlighted by the reviewer regarding the importance of characterizing whether the mobilization of membrane reservoirs and the delivery of additional membrane material are dependent on any cytoskeletal filaments. If so, it will be important to elucidate the mechanisms by which SPI-1 effectors act cooperatively to guide the transport of membrane vesicles. While these questions extend beyond the scope of this current study, we recognize their significance for future investigations. Therefore, we have incorporated additional discussion on this topic in the "Discussion" section (Line 353-355 in the revised manuscript).

REVIEWER COMMENTS

Reviewer #1 (Remarks to the Author):

The authors have done a good job of addressing the reviewers' comments. I still have some minor concerns.

There is inconsistency in how you summarize the roles of the four SPI-1 effectors (SopB, SopD, SopE2, SipC) throughout the paper. In the abstract SopB is the only SPI-1 effector mentioned, where you state that "SopB recruits exocyst subunits from membrane reservoirs and other cellular compartments via independent pathways, thereby allowing exocyst complex assembly and membrane delivery required for ruffle formation and bacterial uptake" (lines 59-62) whereas at the end of the introduction you state "Furthermore, we show that SPI-1 effectors SopB and SopE2 act cooperatively to recruit exocyst subunits from different cellular compartments separately. Together, these independent effector-driven pathways contribute to the formation of invasion ruffles and subsequent uptake of bacteria into host cells" (lines 85- 88). Then in the Results Section, you show that SipC, SopB and SopE2, but not SopD, act cooperatively to recruit the exocyst to invasion sites (lines 235 – 249). Finally, in the Discussion you focus on SopB, SopE2 and SopD and don't mention SipC until the final paragraph when you hypothesize that "the combined activities of SopB, SopE2 and SipC on promoting membrane mobilization and the scission promoting activities of SopB and SopD allow STm to efficiently invade into host cells". Because of this I had trouble following the logic about how these "independent effector-driven pathways contribute to the formation of membrane ruffles" and how their "cooperative actions subvert host membrane trafficking".

Overall, the manuscript needs to be checked for grammar/English language. For example, lines 316-318, 357-359 and 516-519.

Ln 514 Gentamicin not Gentamycin

For % solutions must indicate w/v or v/v.

Ln 604-606 This is confusing. What does "the number was manually measured" mean? Do you mean that cells were scored manually for presence (at least one tubule >10µm) or absence of Rab10+ tubules?

Reviewer #2 (Remarks to the Author):

The authors have sufficiently answered my concerns and questions.

Reviewer #3 (Remarks to the Author):

In the revised version of the manuscript and the explanation in the rebuttal letter, the authors have sufficiently addressed most of my criticism and requests for additional experiments. Most importantly and also requested by reviewer 2, initial evidence has been provided that endogenous Rab10 is present on PM invaginations. Also, the authors did a good job in improving membrane ruffle quantification. A number of questions regarding the mechanisms of SopB function remain open, but for sure this work will inspire future studies.

Specific comments:

However, regarding presence of endogenous Rab10 on PM invaginations, we are left with the just one example of colocalization shown in Fig. 2f, g. The example shown may indeed represent presence of Rab10 on PM-derive tubules, or be a random coincidence. To strengthen this important

point, quantitative data for a sufficient number of cells are required, as well as suitable controls (absence of other Rab not expected on PM tubules, Rab10 k/o, etc.). The alternative approach to demonstrate that the observed phenotypes are independent from Rab10 overexpression could be endogenous GFP tagging by CRISPR/Cas9, but this is likely outside the time frame of the revision.

Reviewer #1 (Remarks to the Author):

The authors have done a good job of addressing the reviewers' comments. I still have some minor concerns.

There is inconsistency in how you summarize the roles of the four SPI-1 effectors (SopB, SopD, SopE2, SipC) throughout the paper. In the abstract SopB is the only SPI-1 effector mentioned, where you state that "SopB recruits exocyst subunits from membrane reservoirs and other cellular compartments via independent pathways, thereby allowing exocyst complex assembly and membrane delivery required for ruffle formation and bacterial uptake" (lines 59-62) whereas at the end of the introduction you state "Furthermore, we show that SPI-1 effectors SopB and SopE2 act cooperatively to recruit exocyst subunits from different cellular compartments separately. Together, these independent effector-driven pathways contribute to the formation of invasion ruffles and subsequent uptake of bacteria into host cells" (lines 85-88). Then in the Results Section, you show that SipC, SopB and SopE2, but not SopD, act cooperatively to recruit the exocyst to invasion sites (lines 235 – 249). Finally, in the Discussion you focus on SopB, SopE2 and SopD and don't mention SipC until the final paragraph when you hypothesize that "the combined activities of SopB, SopE2 and SipC on promoting membrane mobilization and the scission promoting activities of SopB and SopD allow STm to efficiently invade into host cells". Because of this I had trouble following the logic about how these "independent effector-driven pathways contribute to the formation of membrane ruffles" and how their "cooperative actions subvert host membrane trafficking".

Response: We appreciate the reviewer pointing out the confusing language surrounding the role of SPI-1 T3SS effectors. Of the four SPI-I effectors mentioned in the paper, we identified SipC, SopB, and SopE2 as having a role in exocyst recruitment to invasion sites (consistent with prior work by the Casanova group). We have made numerous changes in the manuscript to make the language more consistent, including references to all three of SipC, SopE2, and SopB in the abstract and tightening up the descriptions of the effector's mode of action. Specifically, we refer to the action of SopB, SopE2, and SipC as being cooperative with respect to their ability to recruit the exocyst to invasion sites. The descriptor "independent" was removed when referencing the three effectors together and reserved only for describing the two different SopB-dependent pathways (from RAB10⁺ invaginations and from the Golgi). These changes have removed the inconsistencies in how the roles of these SPI-I effectors are summarized.

Overall, the manuscript needs to be checked for grammar/English language. For example, lines 316-318, 357-359 and 516-519.

Response: The entire manuscript was checked for grammar, especially those lines indicated.

Ln 514 Gentamicin not Gentamycin

Response: The spelling was corrected. We appreciate the thorough reading that caught this error!

For % solutions must indicate w/v or v/v.

Response: We thank the reviewer for the suggestion. We now have included the w/v or v/v information for all % solutions used.

Ln 604-606 This is confusing. What does “the number was manually measured” mean? Do you mean that cells were scored manually for presence (at least one tubule >10µm) or absence of Rab10+ tubules?

Response: Apologies for any previous confusion that may have arisen. The reviewer has interpreted this sentence correctly despite our very confusing language. We have revised this sentence to be clearer and it now reads: For the percentage of cells containing at least one RAB10⁺ tubular membrane reservoir >10 µm in length (tubule length was measured by Fiji), cells were scored manually for the presence or absence of at least one Rab10⁺ tubule over 10 µm and over 100 cells were analyzed in each condition and repeated for three independent experiments.

Reviewer #2 (Remarks to the Author):

The authors have sufficiently answered my concerns and questions.

Response: We genuinely appreciate the reviewer's positive feedback regarding our study. The comments and questions raised by the reviewer have been helpful in enhancing the completeness and significance of our research narrative.

Reviewer #3 (Remarks to the Author):

In the revised version of the manuscript and the explanation in the rebuttal letter, the authors have sufficiently addressed most of my criticism and requests for additional experiments. Most importantly and also requested by reviewer 2, initial evidence has been provided that endogenous Rab10 is present on PM invaginations. Also, the authors did a good job in improving membrane ruffle quantification. A number of questions regarding the mechanisms of SopB function remain open, but for sure this work will inspire future studies.

Specific comments:

However, regarding presence of endogenous Rab10 on PM invaginations, we are left with the just one example of colocalization shown in Fig. 2f, g. The example shown may indeed represent presence of Rab10 on PM-derive tubules, or be a random coincidence. To strengthen this important point, quantitative data for a sufficient number of cells are required, as well as suitable controls (absence of other Rab not expected on PM tubules, Rab10 k/o, etc.). The

alternative approach to demonstrate that the observed phenotypes are independent from Rab10 overexpression could be endogenous GFP tagging by CRISPR/Cas9, but this is likely outside the time frame of the revision.

Response: We are grateful for the thoughtful suggestions and guidance provided by the reviewer.

In our original submission, we presented quantitative evidence demonstrating the localization of myc- or fluorescence-tagged RAB10 to PM-derived tubules. Additionally, we included representative images depicting the colocalization of endogenous RAB10 with a fixable PM marker (PM-mCherry) in last revision.

Here, to further substantiate our conclusions, as per the reviewer's recommendation, we have introduced quantification data that supports the presence of endogenous RAB10 on PM-derived tubules (Supplemental Fig. 2c in the revised manuscript). Moreover, we have undertaken additional experiments involving the staining of endogenous endosomal RAB GTPases. The results confirm the absence of all tested RAB proteins from PM-mCherry⁺ tubules, providing conclusive evidence that RAB10⁺ membrane reservoirs (PM-mCherry⁺) are distinct structures separate from endosomal compartments (Supplemental Fig. 2b,c in the revised manuscript).

Regarding another suggestion from the reviewer that we could employ RAB10 KO cells to assess RAB10's role on PM tubules, we wish to clarify that we have already included pertinent data in the manuscript (Fig 3c,d in the revised manuscript). Our findings demonstrate that RAB10's localized activity on the PM is indeed essential for these tubules.

Overall, we believe that the inclusion of these data strengthens the point that endogenous RAB10 localizes to these PM-derived tubules and plays a vital role for the generation of these structures.